# Momentum Aggregation for Private Non-convex ERM

**Hoang Tran**
Boston University
tranhp@bu.edu

**Ashok Cutkosky**
Boston University
ashok@cutkosky.com

## Abstract

We introduce new algorithms and convergence guarantees for privacy-preserving non-convex Empirical Risk Minimization (ERM) on smooth $d$-dimensional objectives. We develop an improved sensitivity analysis of stochastic gradient descent on smooth objectives that exploits the recurrence of examples in different epochs. By combining this new approach with recent analysis of momentum with private aggregation techniques, we provide an $(\epsilon, \delta)$-differential private algorithm that finds a gradient of norm $\tilde{O}\left(\frac{d^{1/3}}{(\epsilon N)^{2/3}}\right)$ in $O\left(\frac{N^{7/3}\epsilon^{4/3}}{d^{2/3}}\right)$ gradient evaluations, improving the previous best gradient bound of $\tilde{O}\left(\frac{d^{1/4}}{\sqrt{\epsilon N}}\right)$.

## 1 Introduction

In recent years, statistical machine learning models have been deployed in many domains such as health care, education, criminal justice, or social studies [Chen et al., 2021, He et al., 2019, Jiang et al., 2017]. However, the release of statistical estimates based on these sensitive data comes with the risk of leaking personal information of individuals in the original dataset. One naive solution for this problem is to remove all the identifying information such as names, races, or social security numbers. Unfortunately, this is usually not enough to preserve privacy. It has been shown in various works that an adversary can take advantages of structural properties of the rest of the dataset to reconstruct information about certain individuals [Backstrom et al., 2007, Dinur and Nissim, 2003]. Thus, we would need a stronger privacy-preserving mechanism. Over the past couple of decades, differential privacy [Dwork et al., 2006] has emerged as the dominant privacy notion for machine learning problems.

**Definition 1** (Differential Privacy [Dwork and Roth, 2014]). *A randomized algorithm $M: \mathcal{X}^N \mapsto \mathbb{R}^d$ satisfies $(\epsilon, \delta)-$ differential privacy $((\epsilon, \delta)$-DP) if for any two data sets $D, D' \in \mathcal{X}^N$ differing by at most one element and any event $E \subseteq \mathbb{R}^d$, it holds that:*

$$P\left[M(D) \in E\right] \leq \exp(\epsilon)P\left[M(D') \in E\right] + \delta$$

Roughly speaking, differential privacy guarantees that the outputs of two neighboring datasets (datasets that differ in at most one datapoint) are *almost* the same with high probability, thus preventing the adversary from identifying any individual's data.

In this paper, we are interested in designing $(\epsilon, \delta)-$private algorithms for non-convex empirical risk minimization (ERM) problems. In ERM problems, given $N$ i.i.d samples $x_1, ..., x_N \in \mathcal{X}$ from some unknown distribution $P$, the goal is to find $w \in \mathbb{R}^d$ such that $w$ minimizes the empirical loss defined as follows:

$$F(w) \triangleq \frac{1}{N}\sum_{i=1}^{N}f(w, x_i)$$

36th Conference on Neural Information Processing Systems (NeurIPS 2022).

where $f : \mathbb{R}^d \times \mathcal{X} \mapsto \mathbb{R}$ is the loss function associated with the learning problem. This is a setting that commonly arises in modern machine learning problem. For example, in image classification problems, the data point $x$ would be a tuple of (image, label), $w$ denotes the parameters of our model, and $f(w, x)$ represents composing the model predictions with some loss function such as cross-entropy. We are interested in finding a critical point, or a point such that the norm of the empirical gradient $\|\nabla F(w)\|$ is as small as possible. Further, we want all the outputs $w_1, w_2, ..., w_T$ to be differentially private with respect to the $N$ training samples.

Private ERM has been well studied in the convex settings. The approaches in this line of work can be classified into three main categories: *output perturbation* [Dwork et al., 2006, Chaudhuri et al., 2011, Zhang et al., 2017, Wu et al., 2017], *objective perturbation* [Chaudhuri et al., 2011, Kifer et al., 2012, Iyengar et al., 2019, Talwar et al., 2014], and *gradient perturbation* [Bassily et al., 2014, Wang et al., 2017, Jayaraman et al., 2018, Wang et al., 2018]. All of these approaches have been shown to achieve the asymptotically optimal bound $\tilde{O}\left(\frac{\sqrt{d}}{\epsilon N}\right)$ for smooth convex loss (with output perturbation requiring strong convexity to get the optimal bound) in (near) linear time. On the other hand, the literature on private non-convex ERM is nowhere as comprehensive. The first theoretical bound in private non-convex ERM is from [Zhang et al., 2017]. They propose an algorithm called Random Round Private Stochastic Gradient Descent (RRSGD) which is inspired by the results from [Bassily et al., 2014, Ghadimi and Lan, 2013]. RRSGD is able to guarantee the utility bound of $O\left(\frac{(d \log(n/\delta) \log(1/\delta))^{1/4}}{\sqrt{\epsilon N}}\right)$. However, RRSGD takes $O\left(N^2 d\right)$ gradient computations to achieve this, which can be troublesome for high-dimensional problem. [Wang et al., 2018] then improves upon this utility bound by a factor of $O\left((\log(n/\delta))^{1/4}\right)$. They achieve this rate by using full-batch gradient descent which is not a common practice in non-private machine learning in which very large batch sizes actually require careful work to make training efficient. Recently, [Wang et al., 2019b] tackles both runtime and utility issues by introducing a private version of the Stochastic Recursive Momentum (DP-SRM) [Cutkosky and Orabona, 2019]. By appealing to variance reduction as well as privacy amplification by subsampling [Balle et al., 2018, Abadi et al., 2016], DP-SRM achieve the bound $O\left(\frac{(d \log(1/\delta))^{1/4}}{\sqrt{\epsilon N}}\right)$ in $O\left(\frac{(\epsilon N)^{3/2}}{d^{3/4}} + \frac{\epsilon N}{\sqrt{d}}\right)$ gradient complexity. However, DR-SRM still requires the batch size to be $O\left(\frac{\sqrt{\epsilon N}}{d^{1/4}}\right)$ for the analysis to work. Finally, although our focus in this paper is the ERM problem, there are also various works on private *stochastic* non-convex/convex optimization [Bassily et al., 2019, 2021a,b, Feldman et al., 2018, 2020, Zhou et al., 2020, Asi et al., 2021, Wang et al., 2019a, Kulkarni et al., 2021].

**Contributions.** We first provide the analysis for the private version of Normalized SGD (DP-NSGD) [Cutkosky and Mehta, 2020] for unconstrained non-convex ERM. By using the tree-aggregation technique [Chan et al., 2011, Dwork et al., 2010] to compute the momentum privately, we can ensure the privacy guarantee while adding noise of only $\tilde{O}\left(\frac{\sqrt{T}}{\epsilon \sqrt{N}}\right)$ (where $T$ is the total number of iterations). This allows us to achieve the same asymptotic bound $\tilde{O}\left(\frac{d^{1/4}}{\sqrt{\epsilon N}}\right)$ on the expectation of the gradient as [Zhang et al., 2017, Wang et al., 2018, 2019b] without appealing to privacy amplification techniques which is usually required for private SGD to have a good utility guarantee [Abadi et al., 2016]. DP-NSGD also does not require a large batch size as in [Wang et al., 2018, 2019b]; it has utility guarantee for any batch size. This tree-aggregation technique is morally similar to the approach in [Kairouz et al., 2021, Guha Thakurta and Smith, 2013] for online learning. However, unlike in [Kairouz et al., 2021, Guha Thakurta and Smith, 2013], we do not restrict our loss function to be convex and we will also extend tree-aggregation technique to SGD with momentum. Further, we provide a new variant of Normalized SGD that takes advantages of the fact that the gradients of the nearby iterates are close to each other due to smoothness. This new algorithm is able to guarantee an error of $\tilde{O}\left(\frac{d^{1/3}}{(\epsilon N)^{2/3}}\right)$ in $O\left(\frac{N^{7/3} \epsilon^{4/3}}{d^{2/3}}\right)$ gradient computations which, to our knowledge, is the best known rate for private non-convex ERM.

**Organization.** The rest of the paper is organized as follows. In section 2, we define our problem of interest and the assumptions that we make on the problem settings. We also provide some background on Differential Privacy as well as some high-level intuition on tree-aggregation technique. We then formally describe our first private variant of Normalized SGD in section 3 and discuss its privacy guarantee and theoretical utility bound. In section 4, we introduce a novel sensitivity-reduced analysis

for Normalized SGD that allows us to improve upon the utility bound in section 3. Finally, we conclude with a discussion in section 5.

## 2   Preliminaries

**Assumptions.** We define our loss function as $f(w, x) : \mathbb{R}^d \times \mathcal{X} \mapsto \mathbb{R}$ where $x \in \mathcal{X}$ is some sample from the dataset. Throughout the paper, we make the following assumptions on $f(w, x)$. Define a differentiable function $f(.) : \mathbb{R}^d \mapsto \mathbb{R}$ to be $G$-Lipschitz iff $\|\nabla f(y)\| \leq G$ for all $y \in \mathbb{R}^d$, and to be $L$-smooth iff $\|\nabla f(y_1) - \nabla f(y_2)\| \leq L\|y_1 - y_2\|$ where $\|.\|$ is the standard 2-norm unless specified otherwise. We assume that $f(w, x)$ is differentiable, $G$-Lipschitz, and $L$-smooth with probability 1. We also assume that provided the initial point $w_1 \in \mathbb{R}^d$ there exists some upper bound $R$ of $F(w_1)$ or formally: $\sup_{w \in \mathbb{R}^d} F(w_1) - F(w) \leq R$.

**Differential Privacy.** Let $D$ be a dataset containing $N$ datapoints. Then two datasets $D, D'$ are said to be neighbors if $\|D - D'\|_1 = 1$ or in other words, the two datasets differ at exactly one entry. Now we have the definition for sensitivity:

**Definition 2.** *($L_2$-sensitivity) The $L_2$-sensitivity of a function $f(.) : \mathcal{X}^N \mapsto \mathbb{R}^d$ is defined as follows:*

$$\Delta(f) = \sup_{D, D'} \|f(D) - f(D')\|_2$$

In this work, we also make use of Renyi Differential Privacy (RDP) [Mironov, 2017]. RDP is a relaxation of $(\epsilon, \delta)-$DP and can be used to improve the utility bound and do composition efficiently while still guaranteeing privacy.

**Definition 3.** *($(\alpha, \epsilon)$-RDP [Mironov, 2017]) A randomized algorithm $f : \mathcal{X}^N \mapsto \mathbb{R}^d$ is said to have $\epsilon-$RDP of order $\alpha$ if for any neighboring datasets $D, D'$ it holds that:*

$$D_\alpha(f(D)\|f(D')) \leq \epsilon$$

*where $D_\alpha(P\|Q) \triangleq \frac{1}{\alpha-1} \log \mathbb{E}_{x \sim Q} \left( \frac{P(x)}{Q(x)} \right)^\alpha$*

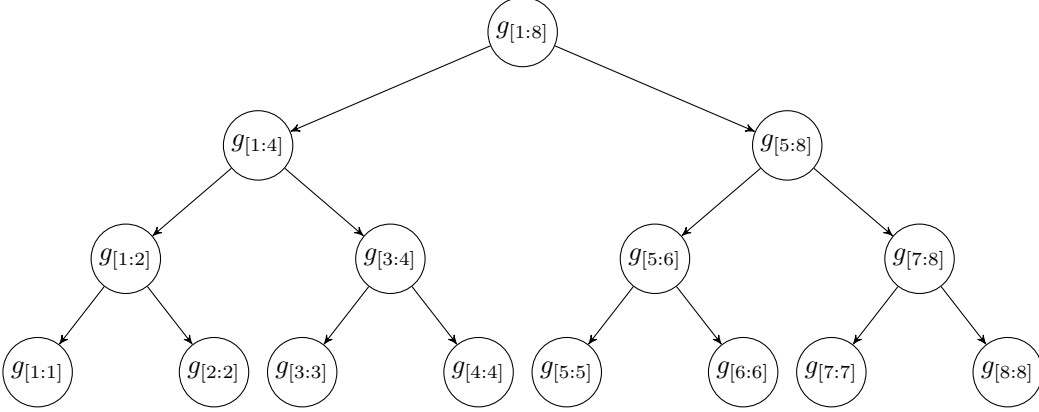

Figure 1: Visualization of the tree-aggregation mechanism. Each node of the tree hold the appropriate value $g_{[y:z]}$ where $g_{[y:z]} = \sum_{i=y}^{z} g_i$.

**Tree-aggregation.** Consider the problem of releasing private partial sum $s_{[y,z]} = \sum_{i=y}^{z} g_i$ given a stream of vectors $g_1, \ldots, g_t$. Assuming $\|g_i\| \leq G \ \forall i \in [t]$, then the simplest solution for this problem is to add Gaussian noise with standard deviation $\tilde{O}\left(1/\epsilon\right)$ to each $g_i$ so that each $g_i$ is $(\epsilon, \delta)-$DP. Then error from the noise added to any partial sum will be bounded by $O(\sqrt{T}/\epsilon)$. However, this problem can be solved much more efficiently using tree-aggregation [Chan et al., 2011, Dwork et al., 2010]. In tree-aggregation, we consider a complete binary tree where each of the leaf nodes is labeled with $g_i$ and each internal node in the tree is the sum of its children. The tree is illustrated in Fig.1. It is clear from Fig.1 that each $g_i$ only affects its ancestors, or at most $\lceil \log t \rceil + 1$ nodes. Thus, by adding Gaussian noise with standard deviation $O(\log t/\epsilon)$ to every node, the complete tree will be $(\epsilon, \delta)-$DP.

Furthermore, since we only need at most $\log t$ nodes to compute any partial sum $s_{[y,z]}$, the noise added to every $s_{[y,z]}$ would be bounded by $O(\log^{1.5} t/\epsilon)$, which is a lot less than the noise that we would add using advanced composition.

To get a general idea on how we can use this tree-aggregation technique to design private algorithms, let us analyse the update of SGD with initial point $w_1 = 0$:

$$w_{t+1} = w_t - \eta \nabla f(w_t, x_t)$$
$$= -\eta \sum_{i=1}^{t} \nabla f(w_i, x_i)$$

where $\nabla f(w_t, x_t)$ is the gradient of the loss function evaluated on the current iterate $w_t$ and some sample $x_t$. Notice that every iterate $w_{t+1}$ would be a function of the sum of the gradients $s_t = \sum_{i=1}^{t} \nabla f(w_t, x_t)$ up to iteration $t$. Thus, if we can use tree-aggregation to privately compute every partial sum $s_{[y,z]} = \sum_{i=y}^{z} \nabla f(w_i, x_i)$, we can the use these partial sums to compute every iterate $w_t$, and $w_t$ is also private by post-processing [Dwork and Roth, 2014]. Similarly, we can also use this aggregation technique for SGD with momentum where we replace $\nabla f(w_t, x_t)$ with the momentum $m_t$. This method of computing momentum would be the main ingredient for our private algorithms. A more detailed discussions are provided in section 3 and 4. Tree-aggregated momentum has been briefly mentioned in the momentum variant of Differentially Private Follow-The-Regularized-Leader (DP-FTRL) in [Kairouz et al., 2021]. However, their algorithm does not directly compose the momentum using the tree-aggregation and they also do not provide any formal guarantee for the algorithm.

## 3   Private Normalized SGD

Before we go into the details of our private algorithm, first let us set up some notations that are frequently used in the rest of the paper. We define the set of subsets $I$ as follows:

$$I = \{[a2^b + 1, (a+1)2^b] | a, b \in \mathbb{Z}, [a2^b + 1, (a+1)2^b] \subset \{1, \ldots, T\}\}$$

Each interval in $I$ corresponds to a node in the tree described above (In Fig.1, the set $I$ is $\{[1, 1], [2, 2], [1, 2], \ldots \}$). Thus, the set of intervals $I$ is essentially a different way to store the binary tree for tree-aggregation mechanism. Then, for every interval $[a, b]$, there exists $O(\log T)$ disjoint subsets $[y_1, z_1], \ldots, [y_n, z_n] \in I$ such that $[a, b] = \bigcup_{i=1}^{n} [y_i, z_i]$ Daniely et al. [2015]. We denote the set of such subsets as COMPOSE$(a, b)$, which can be computed using Algorithm 3. Now we can proceed to the analysis of the private Normalized SGD algorithm described in Algorithm 1.

The main idea of Algorithm 1 is based on two observations. The first observation is $w_t$ is a function of the momentum. Thus, similar to SGD, if we can compute the momentum privately, every iterate $w_1, \ldots, w_T$ would also be private by post-processing. The second observation is that $m_t$ is an exponentially weighted sum of the gradients and can be computed efficiently using tree-aggregation. To see this, notice that the momentum $m_t$ can be written as follows:

$$m_t = (1 - \alpha)m_{t-1} + \alpha \nabla f(w_t, x_{\pi_{r_t}^{q_t}})$$
$$= \alpha \sum_{i=1}^{t} (1 - \alpha)^{t-i} \nabla f(w_i, x_{\pi_{r_i}^{q_i}})$$
$$= \alpha \sum_{[y,z] \in \text{COMPOSE}(1,t)} (1 - \alpha)^{t-z} \sum_{t'=y}^{z} (1 - \alpha)^{z-t'} \nabla f(w_{t'}, x_{\pi_{r_{t'}}^{q_{t'}}})$$

where we denote $x_{\pi_{r_i}^{q_i}}$ as the sample drawn at iteration $i = q_i N + r_i$. Now, if we set $\mathbf{f}_{[y,z]} = \alpha \sum_{t'=y}^{z} (1 - \alpha)^{z-t'} \nabla f(w_{t'}, x_{\pi_{r_{t'}}^{q_{t'}}})$, then:

$$m_t = \sum_{[y,z] \in \text{COMPOSE}(1,t)} (1 - \alpha)^{t-z} \mathbf{f}_{[y,z]}$$

Thus, if we consider $\mathbf{f}_{[y,z]}$ as the value of the node $[y,z]$ in the tree, we can compute $m_t$ using at most $O(\log t)$ nodes from the tree. We will also accumulate the noise in the same way. Assuming $\zeta_{[y,z]}$ as the noise we need to add to node $[y,z]$ to make $\mathbf{f}_{[y,z]}$ private, then the noisy $\hat{m}_t$ is:

$$\hat{m}_t = \sum_{[y,z]\in\text{COMPOSE}(1,t)} (1-\alpha)^{t-z}\mathbf{f}_{[y,z]} + \sum_{[y,z]\in\text{COMPOSE}(1,t)} (1-\alpha)^{t-z}\zeta_{[y,z]}$$

$$= m_t + \sum_{[y,z]\in\text{COMPOSE}(1,t)} (1-\alpha)^{t-z}\zeta_{[y,z]}$$

which is essentially the update of Algorithm 1. Then, as long as $\mathbf{f}_{[y,z]}$ has low sensitivity, the noise we need to add to make $m_t$ private would be small since there are only at most $O(\log t)$ intervals in $\text{COMPOSE}(1,t)$.

---

**Algorithm 1** Differentially private Normalized SGD with momentum (DP-NSGD)

---

**Input:** noise parameter $\sigma$, momentum parameter $\alpha = 1 - \beta$, dataset $(x_1, \ldots, x_N)$, tree depth $R$, Lipschitz constant $G$, loss function $f(w,x)$, noise map Z.
$m_0 \leftarrow \hat{m}_0 \leftarrow 0$.
NOISE $\leftarrow 0$
$V \leftarrow (\min(R, \lfloor \log_2(N) \rfloor) + 1)\frac{T}{N} + \sum_{j=\lfloor \log_2(N) \rfloor + 1}^{R} \lfloor \frac{T}{2^j} \rfloor$.
**for** $b = 0, \ldots, R-1$ **do**
  **for** $a = 0, \ldots, \lfloor T/2^b \rfloor - 1$ **do**
    Sample $\zeta_{[a2^b+1,(a+1)2^b]} \sim N(0, 16\alpha^2 G^2 \sigma^2 V I)$.
    Z$[a2^b + 1, (a+1)2^b] \leftarrow \zeta_{[a2^b+1,(a+1)2^b]}$.
  **end for**
**end for**
**for** $q = 0, \ldots, \lfloor T/N \rfloor$ **do**
  Sample a permutation $\pi_1^q, \ldots, \pi_N^q$ of $\{1, \ldots, N\}$ uniformly at random.
  **for** $i = 1, \ldots, \min\{N, T - qN\}$ **do**
    $t \leftarrow qN + i$.
    Send interval $[1,t]$ to Algorithm 3 to get the set $S = \text{COMPOSE}(1,t)$.
    **for** every interval $S_i \in S$ **do**
      NOISE $\leftarrow$ NOISE $+ (1-\alpha)^{t-(S_i[1])}Z[S_i]$ ( $S_i[1]$ indicates the end index of each interval $S_i$)
    **end for**
    $m_t \leftarrow (1-\alpha)m_{t-1} + \alpha\nabla f(w_t, x_{\pi_i}^q)$.
    $\hat{m}_t \leftarrow m_t + $ NOISE
    $w_{t+1} \leftarrow w_t - \eta\frac{\hat{m}_t}{\|\hat{m}_t\|}$
    NOISE $\leftarrow 0$
  **end for**
**end for**
**return** $w_1, \ldots, w_T$

---

**Theorem 4.** *(Privacy guarantee) Suppose that $f(w,x)$ is $G-$Lipschitz for all $w \in \mathbb{R}^d$, $x \in \mathcal{X}$, Algorithm 1 is $\left(z, \frac{z}{2\sigma^2}\right)$ Renyi-differentially private for all $z$. Consequentially, if $\delta \geq \exp(-\epsilon)$, then with $\sigma \geq \frac{2\sqrt{\log(1/\delta)}}{\epsilon}$, Algorithm 1 is $(\epsilon, \delta)$-differentially private.*

*Proof sketch of Theorem 4.* Denote $t = q_t N + r_t$ for any iteration t for $r_t \in [1, N]$ and $q_t \geq 0$. We define:

$$\mathbf{f}_{[y,z]} = \alpha \sum_{t'=y}^{z} (1-\alpha)^{z-t'}\nabla f(w_{t'}, x_{\pi_{r_{t'}}^{q_{t'}}})$$

$$m_t = \sum_{[y,z]\in\text{COMPOSE}(1,t)} (1-\alpha)^{t-z}\mathbf{f}_{[y,z]}$$

To compute the sensitivity of $\mathbf{f}_{[y,z]}$, recall that the loss function $f$ is $G$-Lipschitz. Further, any given datapoint $x_i$ can contribute at most $\lceil |y - z|/N \rceil$ gradient terms in the summation defining $\mathbf{f}_{[y,z]}$. Thus

$\mathbf{f}_{[y,z]}$ has sensitivity $\Delta = 2\alpha G \sum_{j=0}^{\lceil |y-z|/N \rceil - 1} (1-\alpha)^{jN} \leq 4\alpha G$ for $\alpha \geq \frac{1}{N}$ (Proposition 13). Next, we compute the maximum number of interval that contains any datapoint $x_i$ for $i \in \{1, \ldots, N\}$. Given two neighboring datasets D and D', we use $s$ to indicate the index of the datapoint that is different between the two datasets. We then denote $S_{[y,z]} \subset \{1, ..., N\}$ as the set that contains the indices of the datapoints that contribute to node $[y, z]$ (essentially which elements of the dataset are relevant to the value of node $[y, z]$). For any $i \in \{1, \ldots, N\}$, for any $b \leq \lfloor \log_2(N) \rfloor$, there are at most $\frac{T}{N}$ different $a \in \{0, \ldots, \lfloor T/2^b \rfloor - 1\}$ such that $s \in S_{[a2^b+1,(a+1)2^b]}$ for $b \leq \lfloor \log_2(N) \rfloor$. Further, for any $b > \lfloor \log_2(N) \rfloor$, $i \in S_{[a2^b+1,(a+1)2^b]} = \{1, \ldots, N\}$ for all $a$. Therefore, for any $i$ there are at most $V = \min(R+1, \lfloor \log_2(N) + 1 \rfloor) \frac{T}{N} + \sum_{j=\lfloor \log_2(N)+1 \rfloor}^{R} \lfloor \frac{T}{2^j} \rfloor$ sets $S_i$ such that $i \in S_i$. The privacy guarantee then follows via the ordinary composition properties of Renyi differential privacy. The full proof is included in the appendix. $\square$

Before proving our utility bound, we need the following lemma.

**Lemma 5.** *[Essentially Lemma 2 [Cutkosky and Mehta [2020]] Define:*

$$\hat{\epsilon}_t = \hat{m}_t - \nabla F(w_t)$$

*Suppose $w_1, ..., w_T$ is a sequence of iterates defined by $w_{t+1} = w_t - \eta \frac{\hat{m}_t}{\|\hat{m}_t\|}$ for some arbitrary sequence $\hat{m}_1, ..., \hat{m}_T$. Pick $\hat{w}$ uniformly at random from $w_1, ..., w_T$. Then:*

$$\mathbb{E}\left[\|\nabla F(\hat{w})\|\right] \leq \frac{3\,\mathbb{E}\left[(F(w_1) - F(w_{T+1}))\right]}{2\eta T} + \frac{3L\eta}{4} + \frac{3}{T}\sum_{t=1}^{T}\mathbb{E}[\|\hat{\epsilon}_t\|]$$

Now we are ready to prove the utility guarantee of Algorithm 1.

**Theorem 6.** *(Utility guarantee) Assuming $f(w, x)$ is G-Lipschitz, L-smooth for all $w \in \mathbb{R}^d$, $x \in \mathcal{X}$, and $F(w_1)$ is bounded by R. Then Algorithm 1 with $\eta = \frac{1}{\sqrt{NT}}$, $\alpha = \frac{\epsilon N}{T \log_2 T \sqrt{d \log(1/\delta)}}$, $T = \frac{\epsilon N^2}{\log_2 T \sqrt{d \log(1/\delta)}}$, $\epsilon \leq \frac{T \log_2 T \sqrt{d \log(1/\delta)}}{N}$, and $\hat{w}$ that is pick uniformly random from $w_1, \ldots, w_T$ guarantees:*

$$\mathbb{E}\left[\|\nabla F(\hat{w})\|\right] \leq \frac{(\frac{3}{2}R + 6L\sqrt{\log_2 T \log(1/\delta)} + 12G\sqrt{\log_2 T} \log(1/\delta)^{1/4})d^{1/4}}{\sqrt{\epsilon N}}$$

$$+ \frac{3L(d\log(1/\delta))^{1/4}\sqrt{\log_2 T}}{4\sqrt{\epsilon}N^{3/2}} + \frac{6G}{\sqrt{N}} + \frac{6G\log_2 T \sqrt{d \log(1/\delta)}}{\epsilon N}$$

$$\leq \tilde{O}\left(\frac{d^{1/4}}{\sqrt{\epsilon N}} + \frac{1}{\sqrt{N}}\right)$$

The analysis of Algorithm 1 pretty much follows the analysis of non-private Normalized SGD in Cutkosky and Mehta [2020]. We first bound the error between the noisy momentum and the true empirical risk gradient $\mathbb{E}[\|\hat{m}_t - \nabla F(w_t)\|] \leq \mathbb{E}[\|\hat{m}_t - m_t\|] + \mathbb{E}[\|m_t - \nabla F(w_t)\|]$. The first term is bounded by the standard deviation of the added noise. The second term requires a more delicate analysis due to the use of random reshuffling (RR) in our algorithm (which is a more common practice than sampling in real-world problems). This introduces bias to the gradient estimates. Or formally, if we define $q_i = \lfloor \frac{i}{N} \rfloor$ and $r = i - q_i N$ for every iteration $i \in [T]$, we have $\mathbb{E}[\nabla f(w_i, x_{\pi_{r_i}}^{q_i})] \neq \nabla F(w_i)$. Most of the known analysis of RR is done with regular SGD Mishchenko et al. [2020], Nguyen et al. [2019] since the bias becomes an even bigger problem in SGD with momentum due to the accumulative nature of momentum. To circumvent this problem we rewrite $m_t$ as $\alpha \sum_{i=1}^{t} (1-\alpha)^{t-i} \nabla f(w_{q_i N}, x_{\pi_{r_i}}^{q_i}) + O(\eta N)$ and use the fact that $\mathbb{E}[\nabla f(w_{q_i}, x_{\pi_{r_i}}^{q_i N})] = \nabla F(w_{q_i N})$ where $w_{q_i}$ is the iterate at the beginning of the epoch and $x_{\pi_{r_i}}^{q_i}$ is a datapoint that is sampled *in* that particular epoch (this is true because $w_{q_i N}$ is independent of any data sampled in the epoch). Then, by applying Jensen's inequality and the definition of empirical loss, we have the bound $\mathbb{E}[\|\hat{m}_t - \nabla F(w_t)\|] \leq \tilde{O}\left(\sqrt{\alpha} + \eta N + \frac{\eta}{\alpha} + \frac{\alpha\sqrt{dT}}{\epsilon\sqrt{N}}\right)$. Finally, we can apply Lemma 5 with the appropriate hyperparameters to get the utility bound in Theorem 6.

---

**Algorithm 2** Sensitivity reduced normalized SGD

---

**Input:** Initial Point $w_1$, training set $x_1, \ldots, x_N$, learning rates $\eta$, momentum parameter $\alpha = 1 - \beta$, cross-batch parameter $\gamma$, Lipschitz constant $G$, smoothness constant $L$, noise variance $\sigma^2$.
Set $V_{\leq 2N} = 3 \log_2 N$, $V_{>2N} = 4 \frac{T}{N} \log_2(N)$.
Set $\delta_G = 4G$, $\delta_\Delta = \delta_r = 2\eta N$.
**for** $q = 0, \ldots, \lfloor T/N \rfloor$ **do**
    Sample a permutation $\pi_1^q, \ldots, \pi_N^q$ of $\{1, \ldots, N\}$ uniformly at random.
    **for** $r = 0, \ldots, \min\{N - 1, T - qN\}$ **do**
        $t \leftarrow qN + r_t$
        **for** $k = 0, \ldots, \lfloor \log_2(t) \rfloor$ **do**
            **if** $t \mod 2^k = 0$ **then**
                **if** $t \leq 2N$ **then**
                    Sample $\zeta_t^G \sim N(0, \delta_G^2 \sigma^2 V_{\leq 2N} I)$
                **else**
                    Sample $\zeta_t^G \sim N(0, \delta_G^2 \sigma^2 V_{>2N} I)$
                **end if**
                Sample $\zeta_t^r \sim N(0, \delta_r^2 \sigma^2 V_{>2N} I)$.
                $\hat{F}_{[t-2^k+1, t]}^G = \sum_{i=t-2^k+1}^{t} (1 - \alpha)^{t-i} \nabla f(w_{qN}, x_{\pi_{r_i}^q}) + \zeta_t^G$
                $\hat{F}_{[t-2^k+1, t]}^r = \sum_{i=t-2^k+1}^{t} (1 - \alpha)^{t-i} (\nabla f(w_i, x_{\pi_{r_i}^q}) - \nabla f(w_{qN}, x_{\pi_{r_i}^q})) + \zeta_t^r$
                **if** $t - 2^k + 1 > N$ **then**
                    Sample $\zeta_t^\Delta \sim N(0, \delta_\Delta^2 \sigma^2 V_{>2N} I)$.
                    $\hat{F}_{[t-2^k+1, t]}^\Delta = \sum_{i=t-2^k+1}^{t} (1 - \alpha)^{t-i} (\nabla f(w_{qN}, x_{\pi_{r_i}^q}) - \nabla f(w_{(q-1)N}, x_{\pi_{r_i}^q})) + \zeta_t^\Delta$
                **end if**
            **end if**
        **end for**
        Compute $\hat{m}_t$ using the reconstruction algorithm (Algorithm 4).
        $w_t = w_{t-1} - \eta \frac{\hat{m}_t}{\|\hat{m}_t\|}$
    **end for**
**end for**
Return $w_1, \ldots, w_T$

---

## 4   Sensitivity Reduced Normalized SGD

In this section, we will devise an algorithm that has an asymptotically tighter bound than $\tilde{O}\left(\frac{d^{1/4}}{\sqrt{\epsilon N}}\right)$, which is the best known bound for $(\epsilon, \delta)-$DP non-convex ERM. One natural direction is to use a better algorithm for non-convex optimization than SGD. It is known that in non-convex optimization operating on smooth losses, the optimal rate for SGD is $O(1/N^{1/4})$ Arjevani et al. [2019] while more advanced methods that use variance reduction Fang et al. [2018], Zhou et al. [2018], Tran-Dinh et al. [2019] or Hessian-vectors product Arjevani et al. [2020], Tran and Cutkosky [2021] can achieve the rate $O(1/N^{1/3})$. Thus, one could hope that these methods can improve the bound of private non-convex ERM. Unfortunately, this does not seem to be the case in reality. For example, Wang et al. [2019b] proposes a variant of the variance-reduction based stochastic recursive momentum (STORM) Cutkosky and Orabona [2019] called DP-SRM but their algorithm shows little asymptotic improvement compared to previous known bound. DP-SRM improves the error bound by a factor of $\log(n/\delta)$ but this seems to be the result of sampling Abadi et al. [2016] rather than variance reduction. We would like to explore an alternative direction to improve the error bound. Instead of using a better non-convex optimization methods, we will reuse the Normalized SGD in section 3 but with a tighter privacy analysis that requires us to add less noise than $\tilde{O}\left(\frac{\sqrt{T}}{\epsilon \sqrt{N}}\right)$.

The motivation for our methods comes from full-batch SGD. Consider the following full-batch normalized gradient descent update: $w_{t+1} = w_t - \eta \frac{g_t}{\|g_t\|}$ where $g_t = \nabla F(w_t)$. Notice that (for $t > 1$) $g_t = g_{t-1} + \nabla F(w_t) - \nabla F(w_{t-1})$. Now, given $g_{t-1}$, the sensitivity of $g_t$ can be bounded by the sensitivity of $\nabla F(w_t) - \nabla F(w_{t-1})$, which in turn is only $O(\eta/N)$ due to smoothness and the fact that $\|w_t - w_{t-1}\| = \eta$. This is a lot smaller than the naive bound of $O(1/N)$ since $\eta \ll 1$. We

can iterate the above idea to write $g_t = \nabla F(w_1) + \sum_{i=2}^{t} \nabla F(w_i) - \nabla F(w_{i-1})$. In this way, each $g_t$ is a partial sum of terms with sensitivity $O(\eta/N)$, which can be be privately estimated with an error of $\tilde{O}(\eta/N\epsilon)$ using tree-aggregation. The main intuition here is that except for the first iterate, each iterate is close to the previous iterate due to smoothness - that is, $\nabla F(w_t) - \nabla F(w_{t-1})$ has low sensitivity. In fact, we can further control the sensitivity of $g_t$ by expressing $g_t = (1 - \gamma)(g_{t-1} + \nabla F(w_t) - \nabla F(w_{t-1})) + \gamma \nabla F(w_t)$ for some parameter $\gamma < 1$. Now, with appropriate value of $\gamma$, $g_t$ can potentially have even lower sensitivity compared to the case $\gamma = 0$ that we discuss above. Moving beyond the full-batch case, we would like to replace the full-batch gradients $\nabla F(w_t)$ in this argument with momentum estimates $m_t$. Unfortunately, it is not the case that $m_t - m_{t-1}$ has low sensitivity. Instead, notice that the distance between two iterates evaluated on the same sample at two consecutive epochs will be at most $\eta N \ll 1$ for appropriate $\eta$, and we can show that the sensitivity of $m_t - m_{t-N}$ scales by this same distance factor. Our method described in Algorithm 2 follows this observation to compute low sensitivity momentum queries. To see this, let us rewrite the momentum $m_t$ in Algorithm 1:

$$m_t = (1 - \alpha)m_{t-1} + \alpha \nabla f(w_t, x_{\pi_{r_t}^{q_t}})$$

$$= \alpha \sum_{i=1}^{t} (1 - \alpha)^{t-i} \nabla f(w_i, x_{\pi_{r_i}^{q_i}})$$

$$= \alpha \sum_{i=1}^{t} (1 - \alpha)^{t-i} \nabla f(w_{q_i N}, x_{\pi_{r_i}^{q_i}}) + \alpha \sum_{i=1}^{t} (1 - \alpha)^{t-i} (\nabla f(w_i, x_{\pi_{r_i}^{q_i}}) - \nabla f(w_{q_i N}, x_{\pi_{r_i}^{q_i}}))$$

$$\tag{1}$$

In order to exploit this decomposition, we define $G_{[a,b]} = \sum_{t=a}^{b} (1 - \alpha)^{b-t} \nabla f(w_{q_t N}, x_{\pi_{r_t}^{q_t}})$, $r_{[a,b]} = \sum_{t=a}^{b} (1 - \alpha)^{b-t} (\nabla f(w_t, x_{\pi_{r_t}^{q_t}}) - \nabla f(w_{q_t N}, x_{\pi_{r_t}^{q_t}}))$ and $\Delta_{[a,b]} = \sum_{t=a}^{b} (1 - \alpha)^{b-t} (\nabla f(w_{q_t N}, x_{\pi_{r_t}^{q_t}}) - \nabla f(w_{(q_t-1)N}, x_{\pi_{r_t}^{q_t}}))$. Then, (1) states:

$$m_t = \alpha G_{[1,t]} + \alpha r_{[1,t]} \tag{2}$$

Considering $t = qN + r$, we can further refine this expansion by observing (Lemma 17):

$$G_{[1,t]} = (1 - \alpha)^{t-r} G_{[1,r]} + \sum_{i=0}^{q-1} (1 - \alpha)^{Ni} (1 - \gamma)^{q-(i+1)} G_{[r+1,r+N]}$$

$$+ \sum_{i=1}^{q-1} \left( \sum_{j=0}^{i-1} (1 - \gamma)^j (1 - \alpha)^{(i-1-j)N} \right) \left( (1 - \gamma) \Delta_{[t-iN+1,t-(i-1)N]} + \gamma G_{[t-iN+1,t-(i-1)N]} \right)$$

$$\tag{3}$$

Thus, in Algorithm 2, we will estimate $G_{[a,b]}$, $r_{[a,b]}$ and $\Delta_{[a,b]}$ for all intervals $[a, b]$ using tree aggregation, and then recombine these estimates using (2) and (3) to estimate $m_t$. The critical observation is that each term of the sums for $r_{[a,b]}$ and $\Delta_{[a,b]}$ have sensitivity at most $O(\eta N)$, because the normalized update implies $\|w_t - w_{t'}\| \leq \eta N$ for all $|t - t'| \leq N$, while the $O(1)$ sensitivity terms in $G_{[a,b]}$ are scaled down by $\gamma$. However, notice that in the first two epochs, we don't have the extra parameter $\gamma$ that helps us control the noise added to the momentum. Thus, if we just naively add noise based on the maximum sensitivity (which is $V_{>2N}$), the noise will blow up since it scales with $T/N$. To remedy this, we add two different types of noise to $G_{[t-2^k+1,t]}$ ($k \in \{0, \ldots, \lfloor \log_2(t) \rfloor\}$) depends on whether $t \geq 2N$. By dividing the sensitivity into 2 cases, we can make the momentum of the first 2 epochs $(\epsilon, \delta)$−DP by adding noise of $\tilde{O}(1/\epsilon)$ and every other epochs $(\epsilon, \delta)$−DP by adding noise of $\tilde{O}(\gamma\sqrt{T}/\sqrt{N}\epsilon)$ . Then, the whole procedure would be $(O(\epsilon), O(\delta))$−DP overall.

Morally speaking, we can think of the first two terms of the update of Algorithm 2 (Eq. 3) as the first iterate in the full-batch example with some exponentially weighted factors and the third term as the $(1 - \gamma)(\nabla F(w_t) - \nabla F(w_{t-1})) + \gamma \nabla F(w_t)$ part. Overall, if we set $\gamma = \eta N$, the noise we need to add to make $m_t$ private is $\tilde{O}\left( \frac{\alpha}{\epsilon} + \frac{\alpha\sqrt{\eta T}}{\epsilon} + \frac{\alpha\eta N\sqrt{T}}{\epsilon\sqrt{N}} \right)$, which is smaller than the $\tilde{O}(\frac{\alpha\sqrt{T}}{\epsilon\sqrt{N}})$ noise that we add in Section 3. Formally, we have the following error bound for the momentum.

**Lemma 7.** *Suppose $\alpha \geq \frac{1}{N}$ and $\gamma \leq \frac{1}{2}$, using the update of Algorithm 2, we have:*

$$\mathbb{E}[\|\hat{m}_t - m_t\|] \leq \frac{192 G \alpha \log_2 T \sqrt{d \log(1/\delta)}}{\epsilon} + \frac{128(G+2L)\alpha \sqrt{\eta dT \log(1/\delta)} \log_2 T}{\epsilon}$$
$$+ \frac{16\eta L \alpha \log_2 T \sqrt{3dTN \log(1/\delta)}}{\epsilon}$$

To prove Lemma 7, we sum up the variance of the Gaussian noise needed to add to make every term in Eq.3 and $\hat{r}_t$ private. Then $\mathbb{E}[\|\hat{m}_t - m_t\|] \leq \sqrt{d\text{VAR}}$ where VAR is the total noise variance and $d$ is the dimension of the noise vector.

**Theorem 8.** *(Privacy guarantee) Suppose $\alpha \geq 1/N$ and $f$ is G-Lipschitz and L-smooth. Then Algorithm 2 is $(z, 3z/2\sigma^2)$ Renyi differentially private for all $z$. With $\sigma \geq \frac{4\sqrt{\log(1/\delta)}}{\epsilon}$, Algorithm 2 is $(\epsilon, \delta)$-DP*

The proof of Theorem 8 follows the proof of tree-aggregated momentum with Renyi differential privacy as in section 3. We defer the proof to the appendix.

Now we have the utility bound for Algorithm 2.

**Theorem 9.** *(Utility guarantee) Assuming $f(w, x)$ is G-Lipschitz, L-smooth for all $w \in \mathbb{R}^d$, $x \in \mathcal{X}$, and $F(w_1)$ is bounded by R. Then Algorithm 2 with $\gamma = \eta N, \eta = \frac{1}{\sqrt{NT}}, \alpha = \frac{N^{3/4}\epsilon}{T^{3/4}\sqrt{d}}, T = \frac{N^{7/3}\epsilon^{4/3}}{d^{2/3}}$, $\epsilon \leq \frac{T^{3/4}\sqrt{d}}{N^{3/4}}$, and $\hat{w}$ that is picked uniformly at random from $w_1, ..., w_T$ guarantees:*

$$\mathbb{E}[\|\nabla F(\hat{w})\|] \leq \frac{(\frac{3}{2}R + 24K(G+2L) + 12L)d^{1/3}}{(\epsilon N)^{2/3}} + \frac{6G}{\sqrt{N}} + \frac{36GK\sqrt{d}}{\epsilon N} + \frac{9KL\sqrt{d}}{\epsilon N}$$

$$\leq \tilde{O}\left(\frac{d^{1/3}}{(\epsilon N)^{2/3}} + \frac{1}{\sqrt{N}}\right)$$

*where $K = 16 \log_2 T \log(1/\delta)$.*

The analysis of Algorithm 2 is similar to the analysis of Algorithm 1. We still bound the error between the noisy momentum and the true empirical risk gradient as $\mathbb{E}[\|\hat{m}_t - \nabla F(w_t)\|] \leq \mathbb{E}[\|\hat{m}_t - m_t\|] + \mathbb{E}[\|m_t - \nabla F(w_t)\|]$. $\mathbb{E}[\|m_t - \nabla F(w_t)\|]$ is the same as in Theorem 6 but now, the noise bound $\mathbb{E}[\|\hat{m}_t - m_t\|]$ is only $\tilde{O}\left(\frac{\alpha\sqrt{d}}{\epsilon} + \frac{\alpha\sqrt{\eta dT}}{\epsilon} + \frac{\alpha\eta N\sqrt{dT}}{\epsilon\sqrt{N}}\right)$ instead of $\tilde{O}\left(\frac{\alpha\sqrt{dT}}{\epsilon\sqrt{N}}\right)$ due to our sensitivity-reduced analysis. Then we can apply Lemma 5 to get our final utility guarantee. The full proof is provided in the appendix.

## 5   Conclusions

In this paper, we present two new private algorithms for non-convex ERM. Both algorithms are variants of Normalized SGD with momentum and both utilizes tree-aggregation method to privately compute the momentum. Our first algorithm overcomes the large batch size and privacy amplification techniques requirements in previous works while still achieving the state-of-the-art asymptotic bound $\tilde{O}\left(\frac{d^{1/4}}{\sqrt{\epsilon N}}\right)$. Our second algorithm uses the insight from full-batch SGD operating on smooth losses to reduce the sensitivity of the momentum. This allows the algorithm to achieve the utility bound $\tilde{O}\left(\frac{d^{1/3}}{(\epsilon N)^{2/3}}\right)$, which, to our knowledge, is the best known bound for private non-covex ERM.

**Limitations:** There are several limitations that one could further explore to improve the results of this paper. The most natural direction is to incorporate privacy amplification by shuffling into both algorithms. Based on the current analysis, it is not obvious that we can apply privacy amplification immediately to tree-aggregated momentum. However, it is possible that there are some clever ways that one can come up to make the analysis work. It has been shown in previous work that privacy amplification by shuffling does help us achieve the $\tilde{O}\left(\frac{d^{1/4}}{\sqrt{\epsilon N}}\right)$ utility bound. Therefore, it is our hope that by combining privacy amplification with our sensitivity reduced algorithm, we can achieve an even better utility bound than $\tilde{O}\left(\frac{d^{1/3}}{(\epsilon N)^{2/3}}\right)$. Furthermore, even though our first

algorithm has comparable run time to previous works, the second algorithm has slightly less ideal run time. The current algorithm requires $\tilde{O}\left(\frac{N^{7/3}\epsilon^{4/3}}{d^{2/3}}\right)$ gradient evaluations but has the total run time of $\tilde{O}(N^{11/3}/d^{2/3})$ due to the extra run time from the compose and reconstruction algorithms. We would like to reduce this to quadratic run time. We will leave these for future research.

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
