## A  Compose algorithm

The algorithm will return a list of intervals $S$ and for any $S_i \in S$, $S_i[0]$ is the start and $S_i[1]$ is the end of the respective interval. Intuitively, this COMPOSE function tells us which nodes in the tree that we need to use to compute any partial sum.

---
**Algorithm 3** COMPOSE $[a, b]$

---
   **Input:**  Starting point $a$, ending point $b$ of interval $[a, b]$.
   Let $k$ be the largest $k$ such that $a = 1 \mod 2^k$ and $a + 2^k - 1 \le b$.
   Set $S = \{[a, a + 2^k - 1]\}$
   Set $a' = a + 2^k$.
   **if** $a' > b$ **then**
      **return** $S$.
   **else**
      Let $S' = \text{COMPOSE}(a', b)$.
      Let $S = S \cup S'$.
      **return** $S$
   **end if**

---

## B  Reconstruction algorithm for sensitivity-reduced algorithm (Algorithm 2)

---
**Algorithm 4** Reconstruction algorithm

---
   **Input:**  Noisy gradient arrays $\hat{F}^G, \hat{F}^\Delta, \hat{F}^r$, momentum parameter $\alpha$, sensitivity parameter $\gamma$, iteration $t$.
   $q \leftarrow \lfloor \frac{t}{N} \rfloor$
   $r = t - qN$
   $\hat{G}_{[1,r]} = \sum_{[y,z] \in \text{COMPOSE}(1,r)} (1 - \alpha)^{r-z} \hat{F}^G_{[y,z]}$
   $\hat{G}_{[r+1,r+N]} = \sum_{[y,z] \in \text{COMPOSE}(r+1,r+N)} (1 - \alpha)^{r+N-z} \hat{F}^G_{[y,z]}$
   $\hat{G}_{[1,t]} = (1 - \alpha)^{t-r} \hat{G}_{[1,r]}$ //initial value for $\hat{G}_{[1,t]}$ that will be updated in the following loop.
   **for** $i = 0 \ldots q - 1$ **do**
     $\hat{G}_{[t-iN+1, t-(i-1)N]} = \sum_{[y,z] \in \text{COMPOSE}(t-iN+1, t-(i-1)N)} (1 - \alpha)^{t-(i-1)N-z} \hat{F}^G_{[y,z]}$
     $\hat{\Delta}_{[t-iN+1, t-(i-1)N]} = \sum_{[y,z] \in \text{COMPOSE}(t-iN+1, t-(i-1)N)} (1 - \alpha)^{t-(i-1)N-z} \hat{F}^\Delta_{[y,z]}$
     $\hat{G}_{[1,t]} += (\sum_{j=0}^{i-1} (1-\gamma)^j (1-\alpha)^{(i-1-j)N})((1-\gamma)\hat{\Delta}_{t-iN+1,t-(i-1)N} + \gamma \hat{G}_{t-iN+1,t-(i-1)N}) +$
     $(1-\alpha)^{Ni} (1-\gamma)^{q-i-1} \hat{G}_{[r+1,r+N]}$
   **end for**
   $\hat{r}_t = \sum_{[y,z] \in \text{COMPOSE}(1,t)} (1 - \alpha)^{t-z} \hat{F}^r_{[y,z]}$
   $\hat{m}_t = \alpha \hat{G}_{[1,t]} + \alpha \hat{r}_t$
   **Return:**  $\hat{m}_t$

---

## C  Renyi Differential Privacy

In this section, we will prove some general theorems on the composition of RDP using tree compositions. Note that these results are all consequences of well-known properties of Renyi differential privacy [Mironov, 2017] and tree aggregation [Dwork et al., 2010, Chan et al., 2011] and have been used in many other settings [Guha Thakurta and Smith, 2013, Kairouz et al., 2021, Asi et al., 2021]. However, we find many presentations lacking in some detail, so we reproduce a complete description and analysi here for completeness.

We will consider functions operating on datasets of size $N$. Given two neighboring datasets $D = (Z_1, \ldots, Z_N)$ and $D' = (Z'_1, \ldots, Z'_N)$, we use $s$ to indicate the index that is different between the datasets. That is, $Z_i = Z'_i$ for $i \ne s$. Given a subset $S \subset \{1, \ldots, N\}$, we use $D[S]$ to indicate the restriction of $D$ to the elements with index in $S$: $D[S] = (Z_i \mid i \in S)$.

Consider a set of $K$ functions $G_1, \ldots, G_K$, and an associated set of *subsets* of $\{1, \ldots, N\}$, $S_1, \ldots, S_K$ (note that the $S_i$ are sets of integers, NOT sets of datapoints). Each $G_i$ produces outputs in a space $\mathcal{W}$, and takes $i$ inputs: first, a dataset $D$ (or $D'$) of size $N$, and then $i - 1$ elements of $\mathcal{W}$. That is, if the space of all datasets of size $N$ is $\mathcal{D}$, then $G_i : \mathcal{D} \times \mathcal{W}^{i-1} \to \mathcal{W}$. Further, each $G_i$ must have the property that $G_i$ depends *only* on $D[S_i]$. that is, if $q \notin S_i$, then $G_i(D, x_1, \ldots, x_{i-1}) = G_i(D', x_1, \ldots, x_{i-1})$ for all $x_1, \ldots, x_{i-1}$. With this in mind, we recursively define:

$$\begin{aligned}
\mathbf{f}_1 &= G_1(D) \\
\mathbf{f}_1' &= G_1(D') \\
\mathbf{f}_i &= G_i(D, \mathbf{f}_1, \ldots, \mathbf{f}_{i-1}) \\
\mathbf{f}_i' &= G_i(D', \mathbf{f}_1', \ldots, \mathbf{f}_{i-1}')
\end{aligned}$$

Thus, the ordering of the $G_i$ indicates a kind of "causality" direction: later $\mathbf{f}_i$ are allowed to depend on the earlier $\mathbf{f}_i$, but the dependencies on the *dataset* are fixed by the $S_i$. Intuitively, we should think of being able to "break up" a set of desired computational problems into a number of smaller computations such that (1) each sub-computation is in some way "local" in that it depends on only a small part of the dataset, potentially given the outputs of previous sub-computations, and (2) each of the original desired computational problems can be recovered from the answers to a small number of the sub-computations. If so, then we will be able to accurately and privately make all the desired computations.

Given the above background, we define:

- $\Delta_i$ to be the maximum sensitivity (with respect to some problem-specific metric like $L_2$ or $L_1$) over all $(x_1, \ldots, x_{i-1}) \in (\mathcal{W})^{i-1}$ of the function $D \mapsto G_i(D, x_1, \ldots, x_{i-1})$. That is, when $\mathcal{W} = \mathbb{R}^d$ and the metric is the one induced by a norm $\| \cdot \|$:

$$\Delta_i = \sup_{|D-D'|=1, x_1, \ldots, x_{i-1}} \|G_i(D, x_1, \ldots, x_{i-1}) - G_i(D', x_1, \ldots, x_{i-1})\|$$

where we use $|\mathcal{D} - \mathcal{D}'| = 1$ to indicate that datasets are neighboring.

- $\text{IN}(s)$ to be the set of indices $i \in \{1, \ldots, N\}$ such that $s \in S_i$.

- $\text{OUT}(s)$ to be the complement of $\text{IN}(s)$: the set of indices $i \in \{1, \ldots, N\}$ such that $s \notin S_i$.

## C.1 Algorithm and Analysis

Now, we will describe the aggregation algorithm that is used to compute private versions of all $f_t$, assuming that $\mathcal{W} = \mathbb{R}^d$ for some $d$. We write $X \sim \mathcal{N}(\mu, \sigma^2)$ to indicate that $X$ has density $p(X = x) = \frac{1}{\sigma\sqrt{2\pi}} \exp\left(-(x - \mu)^2/2\sigma^2\right)$. Further, for a vector $\mu$, we write $X \sim \mathcal{N}(\mu, \sigma^2 I)$ to indicate $P(X = x) = \frac{1}{(\sigma\sqrt{2\pi})^d} \exp\left(-\|x - \mu\|^2/2\sigma^2\right)$

Let $D_\alpha(P\|Q)$ indicate the Renyi divergence between $P$ and $Q$:

$$\begin{aligned}
D_\alpha(P\|Q) &= \frac{1}{\alpha - 1} \log \mathop{\mathbb{E}}_{x \sim Q} \left[ \left( \frac{P(x)}{Q(x)} \right)^\alpha \right] \\
&= \frac{1}{\alpha - 1} \log \left( \int_x Q(x)^{1-\alpha} P(x)^\alpha \, dx \right)
\end{aligned}$$

Now, we need the following fact about Gaussian divergences:

$$D_\alpha(\mathcal{N}(0, \sigma^2)\|\mathcal{N}(\mu, \sigma^2)) = \alpha\mu^2/2\sigma^2$$

This implies the following multi-dimensional extension:

$$D_\alpha(\mathcal{N}(0,\sigma^2 I)\|\mathcal{N}(\mu,\sigma^2 I)) = \frac{1}{\alpha-1}\log\left(\frac{1}{(\sigma\sqrt{2\pi})^d}\int_{\mathbb{R}^d}\exp(-\|x-\mu\|^2/2\sigma^2)^{1-\alpha}\exp(-\|x\|^2/2\sigma^2)^\alpha)\,dx_1\ldots dx_d\right)$$

$$\frac{1}{\alpha-1}\left[\sum_{i=1}^{d}\log\left(\frac{1}{\sigma\sqrt{2\pi}}\int_{-\infty}^{\infty}\exp(-(x-\mu_i)^2/2\sigma^2)^{1-\alpha}\exp(-x^2/2\sigma^2)^\alpha)dx\right)\right]$$

$$=\sum_{i=1}^{d}D_\alpha(\mathcal{N}(0,\sigma^2)\|\mathcal{N}(\mu_i,\sigma^2))$$

$$=\alpha\|\mu\|^2/2\sigma^2$$

---

**Algorithm 5** Aggregation Algorithm with Gaussian noise

---

**Input:** Dataset $D$, functions $G_1,\ldots,G_K$ with sensitivities $\Delta_1,\ldots,\Delta_K$ with respect to the $L_2$ norm. Noise parameter $\rho$
Sample random $\zeta_1 \sim \mathcal{N}(0,\Delta_1^2/\rho^2 I)$
Set $\mathbf{f}_1 = G_1(D)$.
Set $\hat{\mathbf{f}}_1 = \mathbf{f}_1 + \zeta_1$
**for** $i = 2,\ldots,K$ **do**
    Sample random $\zeta_t \sim \mathcal{N}(0,\Delta_i^2/\rho^2 I)$.
    Set $\mathbf{f}_i = G_i(D,\hat{\mathbf{f}}_1,\ldots,\hat{\mathbf{f}}_{i-1})$.
    Set $\hat{\mathbf{f}}_i = \mathbf{f}_i + \zeta_i$.
**end for**
**return** $\hat{\mathbf{f}}_1,\ldots,\hat{\mathbf{f}}_K$.

---

**Theorem 10.** *Let $V$ to be the maximum over all $s \in \{1,\ldots,N\}$ of the total number of sets $S_i$ such that $s \in S_i$ (i.e. $V = \sup_s |\text{IN}(s)|$). Then Algorithm 5 is $(\alpha, V\alpha\rho^2/2)$ Renyi differentially private for all $\alpha$.*

To convert the above result to ordinary differential privacy, we observe that $(\alpha,\epsilon)$-RDP implies $(\epsilon + \frac{\log(1/\delta)}{\alpha-1},\delta)$-DP for all $\delta$. Thus, supposing $\rho \leq \sqrt{\frac{\log(1/\delta)}{V}}$, we then set $\alpha = 1 + \frac{\sqrt{\log(1/\delta)}}{\rho\sqrt{V}}$ to get $(V\rho^2 + \rho\sqrt{V\log(1/\delta)},\delta) \leq (2\rho\sqrt{V\log(1/\delta)},\delta)$ differential privacy.

*Proof.* Let us write $\hat{\mathbf{f}}_i$ for the outputs with input dataset $D$, and $\hat{\mathbf{f}}_i'$ for the outputs with input dataset $D'$. Let $s$ be the index such that $Z_q \neq Z_q'$.

Then, we can express the joint density of the random variable $\hat{\mathbf{f}}_1,\ldots,\hat{\mathbf{f}}_K$:

$$p(\hat{\mathbf{f}}_1 = r_1,\ldots,\hat{\mathbf{f}}_k = r_k) = \prod_{i=1}^{K}p(\hat{\mathbf{f}}_i = r_i|\hat{\mathbf{f}}_1 = r_1,\ldots,\hat{\mathbf{f}}_{i-1} = r_{i-1})$$

$$= \prod_{i\in\text{IN}(s)}p(\hat{\mathbf{f}}_i = r_i|\hat{\mathbf{f}}_1 = r_1,\ldots,\hat{\mathbf{f}}_{i-1} = r_{i-1})\prod_{i\in\text{OUT}(s)}p(\hat{\mathbf{f}}_i = r_i|\hat{\mathbf{f}}_1 = r_1,\ldots,\hat{\mathbf{f}}_{i-1} = r_{i-1})$$

Similar expressions hold for $\hat{\mathbf{f}}_i'$:

$$p(\hat{\mathbf{f}}_1' = r_1,\ldots,\hat{\mathbf{f}}_k' = r_k) = \prod_{i\in\text{IN}(s)}p(\hat{\mathbf{f}}_i' = r_i|\hat{\mathbf{f}}_1' = r_1,\ldots,\hat{\mathbf{f}}_{i-1}' = r_{i-1})\prod_{i\in\text{OUT}(s)}p(\hat{\mathbf{f}}_i' = r_i|\hat{\mathbf{f}}_1' = r_1,\ldots,\hat{\mathbf{f}}_{i-1}' = r_{i-1})$$

Further, for any $i \in \text{OUT}(s)$

$$p(\hat{\mathbf{f}}_i = r_i|\hat{\mathbf{f}}_1 = r_1,\ldots,\hat{\mathbf{f}}_{i-1} = r_{i-1}) = p(\hat{\mathbf{f}}_i' = r_i|\hat{\mathbf{f}}_1' = r_1,\ldots,\hat{\mathbf{f}}_{i-1}' = r_{i-1})$$

Now, for the rest of the proof, we mimic the proof of composition for Renyi differential privacy: let $P$ and $P'$ be the distributions of the ouputs under $D$ and $D'$. Then:

$$D_\alpha(P'||P) = \frac{1}{\alpha - 1} \log \left[ \int_r \prod_{i \in \text{IN}(s)} p(\hat{\mathbf{f}}_i = r_i | \hat{\mathbf{f}}_1 = r_1, \ldots, \hat{\mathbf{f}}_{i-1} = r_{i-1})^{1-\alpha} \right.$$

$$\prod_{i \in \text{OUT}(s)} p(\hat{\mathbf{f}}_i = r_i | \hat{\mathbf{f}}_1 = r_1, \ldots, \hat{\mathbf{f}}_{i-1} = r_{i-1})^{1-\alpha}$$

$$\left. \prod_{i \in \text{IN}(s)} p(\hat{\mathbf{f}}'_i = r_i | \hat{\mathbf{f}}'_1 = r_1, \ldots, \hat{\mathbf{f}}'_{i-1} = r_{i-1})^{\alpha} \prod_{i \in \text{OUT}(s)} p(\hat{\mathbf{f}}'_i = r_i | \hat{\mathbf{f}}'_1 = r_1, \ldots, \hat{\mathbf{f}}'_{i-1} = r_{i-1})^{\alpha} \, dr \right]$$

$$= \frac{1}{\alpha - 1} \log \left[ \int_r \prod_{i \in \text{IN}(s)} p(\hat{\mathbf{f}}_i = r_i | \hat{\mathbf{f}}_1 = r_1, \ldots, \hat{\mathbf{f}}_{i-1} = r_{i-1})^{1-\alpha} p(\hat{\mathbf{f}}'_i = r_i | \hat{\mathbf{f}}'_1 = r_1, \ldots, \hat{\mathbf{f}}'_{i-1} = r_{i-1})^{\alpha} \right.$$

$$\left. \prod_{i \in \text{OUT}(s)} p(\hat{\mathbf{f}}_i = r_i | \hat{\mathbf{f}}_1 = r_1, \ldots, \hat{\mathbf{f}}_{i-1} = r_{i-1})^{1-\alpha} p(\hat{\mathbf{f}}'_i = r_i | \hat{\mathbf{f}}'_1 = r_1, \ldots, \hat{\mathbf{f}}'_{i-1} = r_{i-1})^{\alpha} \, dr \right]$$

To make notation a little more precise, let us write $dr_{\text{IN}}$ to indicate $\wedge_{i \in \text{IN}(s)} dr_i$ and $dr_{\text{OUT}} = \wedge_{i \in \text{OUT}(s)} dr_i$. Similarly, $\int_{r_{IN}}$ indicates integration over only $r_i$ such that $i \in \text{IN}(s)$. Now, recall that for $i \in \text{OUT}(s)$, we have

$$p(\hat{\mathbf{f}}_i = r_i | \hat{\mathbf{f}}_1 = r_1, \ldots, \hat{\mathbf{f}}_{i-1} = r_{i-1}) = p(\hat{\mathbf{f}}'_i = r_i | \hat{\mathbf{f}}'_1 = r_1, \ldots, \hat{\mathbf{f}}'_{i-1} = r_{i-1})$$

so that:

$$D_\alpha(P'||P) = \frac{1}{\alpha - 1} \log \left[ \int_r \prod_{i \in \text{IN}(s)} p(\hat{\mathbf{f}}_i = r_i | \hat{\mathbf{f}}_1 = r_1, \ldots, \hat{\mathbf{f}}_{i-1} = r_{i-1})^{1-\alpha} p(\hat{\mathbf{f}}'_i = r_i | \hat{\mathbf{f}}'_1 = r_1, \ldots, \hat{\mathbf{f}}'_{i-1} = r_{i-1})^{\alpha} \right.$$

$$\left. \prod_{i \in \text{OUT}(s)} p(\hat{\mathbf{f}}_i = r_i | \hat{\mathbf{f}}_1 = r_1, \ldots, \hat{\mathbf{f}}_{i-1} = r_{i-1}) \, dr_{\text{OUT}} dr_{\text{IN}} \right]$$

so, we can integrate over $r_i$ for $i \in \text{OUT}(s)$:

$$= \frac{1}{\alpha - 1} \log \left[ \int_{r_{\text{IN}}} \prod_{i \in \text{IN}(s)} p(\hat{\mathbf{f}}_i = r_i | \hat{\mathbf{f}}_1 = r_1, \ldots, \hat{\mathbf{f}}_{i-1} = r_{i-1})^{1-\alpha} p(\hat{\mathbf{f}}'_i = r_i | \hat{\mathbf{f}}'_1 = r_1, \ldots, \hat{\mathbf{f}}'_{i-1} = r_{i-1})^{\alpha} \, dr_{\text{IN}} \right]$$

Now, let the indices in $\text{IN}(s)$ be (in order): $i_1, \ldots, i_n$. Then we have:

$$D_\alpha(P'||P)$$

$$= \frac{1}{\alpha - 1} \log \left[ \left( \prod_{j=1}^{n} \int_{r_{i_j}} p(\hat{\mathbf{f}}_{i_j} = r_{i_j} | \hat{\mathbf{f}}_1 = r_1, \ldots, \hat{\mathbf{f}}_{i_j - 1} = r_{i_j - 1})^{1-\alpha} p(\hat{\mathbf{f}}'_{i_j} = r_{i_j} | \hat{\mathbf{f}}'_1 = r_1, \ldots, \hat{\mathbf{f}}'_{i_j - 1} = r_{i_j - 1})^{\alpha} \right) dr_{\text{IN}} \right]$$

Now, let's compress the density notation a bit to save space:

$$= \frac{1}{\alpha - 1} \log \left[ \left( \prod_{j=1}^{n} \int_{r_{i_j}} p(\hat{\mathbf{f}}_{i_j} = r_{i_j} | r_1, \ldots, r_{i_j - 1})^{1-\alpha} p(\hat{\mathbf{f}}'_{i_j} = r_{i_j} | r_1, \ldots, r_{i_j - 1})^{\alpha} \right) dr_{i_1} \ldots dr_{i_n} \right]$$

Now, let's focus on just one integral:

$$\int_{r_{i_j}} p(\hat{\mathbf{f}}_{i_j} = r_{i_j} | r_1, \ldots, r_{i_j - 1})^{1-\alpha} p(\hat{\mathbf{f}}'_{i_j} = r_{i_j} | r_1, \ldots, r_{i_j - 1})^{\alpha} dr_{i_j}$$

$$= \int_{r_{i_j}} p(\zeta_{i_j} = r_{i_j} - G_{i_j}(D, r_1, \ldots, r_{i_j - 1}))^{1-\alpha} p(\zeta'_{i_j} = r_{i_j} - G_{i_j}(D', r_1, \ldots, r_{i_j - 1}))^{\alpha} \, dr_{i_j}$$

$$= \frac{1}{(\sigma\sqrt{2\pi})^d} \int_{\mathbb{R}^d} \exp\left( -\frac{(1-\alpha)\|x - G_{i_j}(D, r_1, \ldots, r_{i_j} - 1)\|^2}{\Delta_{i_j}^2 / \rho^2} \right) \exp\left( -\frac{\alpha\|x - G_{i_j}(D', r_1, \ldots, r_{i_j} - 1)\|^2}{\Delta_{i_j}^2 / \rho^2} \right) dx$$

using a change of variables $z = x - G_{i_j}(D', r_1, \ldots, r_{i_j} - 1)$:

$$= \frac{1}{(\sigma\sqrt{2\pi})^d} \int_{\mathbb{R}^d} \exp\left(-\frac{(1-\alpha)\|z - (G_{i_j}(D, r_1, \ldots, r_{i_j} - 1) - G_{i_j}(D', r_1, \ldots, r_{i_j} - 1))\|^2}{\Delta_{i_j}^2/\rho^2}\right) \exp\left(-\frac{\alpha\|z\|^2}{\Delta_{i_j}^2/\rho^2}\right) dz$$

$$= \exp((\alpha - 1)D_\alpha(\mathcal{N}(0, \Delta_{i_j}^2/\rho^2), \mathcal{N}(G_{i_j}(D, r_1, \ldots, r_{i_j} - 1) - G_{i_j}(D', r_1, \ldots, r_{i_j} - 1), \Delta_{i_j}^2/\rho^2)))$$

use our expression for divergence between Gaussians with the same covariance:

$$= \exp\left((\alpha - 1)\alpha\rho^2\|G_{i_j}(D, r_1, \ldots, r_{i_j} - 1) - G_{i_j}(D', r_1, \ldots, r_{i_j} - 1)\|^2/2\Delta_{i_j}^2\right)$$

$$\leq \exp\left((\alpha - 1)\alpha\rho^2/2\right)$$

Now, returning to our bound on the divergence:

$$D_\alpha(P'||P) = \frac{1}{\alpha - 1} \log\left[\left(\prod_{j=1}^n \int_{r_{i_j}} p(\hat{\mathbf{f}}_{i_j} = r_{i_j}|r_1, \ldots, r_{i_j-1})^{1-\alpha} p(\hat{\mathbf{f}}'_{i_j} = r_{i_j}|r_1, \ldots, r_{i_j-1})^\alpha\right) dr_{i_1} \ldots dr_{i_n}\right]$$

rewrite a bit for clarity:

$$= \frac{1}{\alpha - 1} \log\left[\left(\prod_{j=1}^{n-1} \int_{r_{i_j}} p(\hat{\mathbf{f}}_{i_j} = r_{i_j}|r_1, \ldots, r_{i_j-1})^{1-\alpha} p(\hat{\mathbf{f}}'_{i_j} = r_{i_j}|r_1, \ldots, r_{i_j-1})^\alpha\right.\right.$$

$$\left.\left.\int_{r_{i_n}} p(\hat{\mathbf{f}}_{i_n} = r_{i_n}|r_1, \ldots, r_{i_n-1})^{1-\alpha} p(\hat{\mathbf{f}}'_{i_n} = r_{i_n}|r_1, \ldots, r_{i_n-1})^\alpha\right) dr_{i_1} \ldots dr_{i_n}\right]$$

integrate out $r_{i_n}$:

$$= \frac{1}{\alpha - 1} \log\left[\exp\left((\alpha - 1)\alpha\rho^2/2\right)\left(\prod_{j=1}^{n-1} \int_{r_{i_j}} p(\hat{\mathbf{f}}_{i_j} = r_{i_j}|r_1, \ldots, r_{i_j-1})^{1-\alpha} p(\hat{\mathbf{f}}'_{i_j} = r_{i_j}|r_1, \ldots, r_{i_j-1})^\alpha\right)\right.$$

$$\left.dr_{i_1} \ldots dr_{i_{n-1}}\right]$$

now integrate out all the other variables one by one:

$$\leq \frac{1}{\alpha - 1} \log\left[\prod_{j=1}^n \exp\left((\alpha - 1)\alpha\rho^2/2\right)\right]$$

$$= n\alpha\rho^2/2$$

$$\leq V\alpha\rho^2/2$$

$\square$

# D   Proof of section 3

## D.1   Privacy

**Theorem 4.** *(Privacy guarantee) Suppose that $f(w, x)$ is $G-$Lipschitz for all $w \in \mathbb{R}^d$, $x \in \mathcal{X}$, Algorithm 1 is $\left(z, \frac{z}{2\sigma^2}\right)$ Renyi-differentially private for all $z$. Consequentially, if $\delta \geq \exp(-\epsilon)$, then with $\sigma \geq \frac{2\sqrt{\log(1/\delta)}}{\epsilon}$, Algorithm 1 is $(\epsilon, \delta)$-differentially private.*

To see the $(\epsilon, \delta)$-DP result from the RDP bound, we observe that $(z, \frac{z}{2\sigma^2})$-RDP implies $(\frac{z}{2\sigma^2} + \frac{\log(1/\delta)}{z-1}, \delta)$-DP for all $\delta$. Thus, optimizing over $z$, we set $z = 1 + \sqrt{2\sigma^2 \log(1/\delta)}$ to obtain $(\epsilon, \delta)$-DP with $\epsilon = 1/2\sigma^2 + 2\sqrt{\log(1/\delta)/2\sigma^2}$. Thus, by quadratic formula, we ensure $(\epsilon, \delta)$-DP for all $\sigma$ satisfying:

$$\sigma \geq \frac{1}{\sqrt{2\log(1/\delta) + 2\epsilon} - \sqrt{2\log(1/\delta)}}$$

In pursuit of a simpler expression, observe that $\sqrt{x+y} \geq \sqrt{x} + \frac{y}{2\sqrt{x+y}}$, so that it suffices to choose:

$$\sigma \geq \frac{\sqrt{2\log(1/\delta) + 2\epsilon}}{\epsilon}$$

So, in particular if $\delta \geq \exp(-\epsilon)$, then we obtain the expression in the Theorem statement.

*Proof.* Let $t = q_t N + r_t$. Define:

$$\mathbf{f}_{[y,z]} = \alpha \sum_{t=y}^{z} (1-\alpha)^{z-t} \nabla f(w_t, x_{\pi_{\pi_{r_t}}^{q_t}})$$

To compute the sensitivity of $\mathbf{f}_{[y,z]}$, recall that $\|\nabla f(w_t, x_{\pi_{\pi_{r_t}}^{q_t}})\| \leq G$ for all $w$ and $x$. Further, any given $x_i$ can contribute at most $\lceil |y - z|/N \rceil$ gradient terms in the summation defining $\mathbf{f}_{[y,z]}$, and the $j$th such term is scaled by $\alpha(1-\alpha)^{jN}$. Thus, $\mathbf{f}_{[y,z]}$ has sensitivity $\Delta_{[y,z]} = 2\alpha G \sum_{j=0}^{\lceil|z-y|/N\rceil-1}(1-\alpha)^{jN} \leq 4\alpha G$ (by proposition 13) for all $[y, z]$ and $\alpha \geq \frac{1}{N}$.

Next we compute the maximum value over all $s \in \{1, \ldots, N\}$ of $|\{S_i \mid s \in S_i\}|$ (the maximum number of intervals that one index s can belong to). For any $s \in \{1, \ldots, N\}$, for any $b \leq \lfloor \log_2(N) \rfloor$, there are at most $\frac{T}{N}$ different $a \in \{0, \ldots, \lfloor T/2^b \rfloor - 1\}$ such that $s \in S_{[a2^b+1,(a+1)2^b]}$ for $b \leq \lfloor \log_2(N) \rfloor$. Further, for any $b > \lfloor \log_2(N) \rfloor$, $s \in S_{[a2^b+1,(a+1)2^b]} = \{1, \ldots, N\}$ for all $a$. Therefore, for any $s$ there are at most $V = \min(R+1, \lfloor \log_2(N)+1 \rfloor)\frac{T}{N} + \sum_{j=\lfloor\log_2(N)+1\rfloor}^{R}\lfloor\frac{T}{2^j}\rfloor$ sets $S_i$ such that $s \in S_i$.

Now, we show that Algorithm 1 is actually providing output distributed in the same way as the aggregation mechanism in Algorithm 5. To do this, observe that given values for $w_1, \ldots, w_T$, $\hat{\mathbf{f}}_{[y,z]} = \zeta_{[y,z]} + \alpha \sum_{t=y}^{z}(1-\alpha)^{z-t}\nabla f(w_t, x_{\pi_{\pi_{r_t}}^{q_t}})$ where $\zeta_{[y,z]} \sim \mathcal{N}(0, \Delta_{[y,z]}^2 \sigma^2 V I)$ using the notation of Algorithm 5. Then the output of Algorithm 5 is

$$\hat{G}_{[1,t]}(\hat{\mathbf{f}}_{[1,1]}, \ldots, \hat{\mathbf{f}}_{[1,t]}) = \sum_{[y,z]\in\text{COMPOSE}(1,t)} (1-\alpha)^{t-z}\hat{\mathbf{f}}_{[y,z]}$$

$$= \alpha \sum_{[y,z]\in\text{COMPOSE}(1,t)} (1-\alpha)^{t-z} \sum_{t'=y}^{z}(1-\alpha)^{z-t'}\nabla f(w_{t'}, x_{\pi_{\pi_{r_{t'}}}^{q_{t'}}})$$

$$+ \sum_{[y,z]\in\text{COMPOSE}(1,t)} (1-\alpha)^{t-z}\zeta_{[y,z]}$$

$$= \alpha \sum_{t'=1}^{t}(1-\alpha)^{t-t'}\nabla f(w_{t'}, x_{\pi_{\pi_{r_{t'}}}^{q_{t'}}})$$

$$+ \sum_{[y,z]\in\text{COMPOSE}(1,t)} (1-\alpha)^{t-z}\zeta_{[y,z]} \qquad (4)$$

Now, observe that the value of $m_t$ as described in Algorithm 1 can be written as:

$$m_t = (1-\alpha)m_{t-1} + \alpha\nabla f(w_t, x_{\pi_{\pi_{r_t}}^{q_t}})$$

$$= \alpha \sum_{t'=1}^{t}(1-\alpha)^{t-t'}\nabla f(w_{t'}, x_{\pi_{\pi_{r_{t'}}}^{q_{t'}}})$$

Thus our momentum $m_t$ in Algorithm 1 is exactly as the first term in the output of Algorithm 5 (Eq. 4) and NOISE$_t$ is the same as the second term. Then, by Theorem 10, Algorithm 1 is $\left(z, \frac{z}{2\sigma^2}\right)-$RDP for all $z$. $\qquad \square$

## D.2   Utility

To prove the main theorem of section 3 (Theorem 6) we would need some extra lemmas on the momentum error below. Lemma 11 is the bound on the error of $m_t$ without any added noise from the tree. This error comes from the biasedness of the momentum as well as shuffling. Then, we will prove Lemma 12 which is the bound on the added noise.

**Lemma 11.** *Define $m_t$ as:*

$$m_t = (1 - \alpha)m_{t-1} + \alpha \nabla f(w_t, x_{\pi_{r_t}^{q_t}})$$

*where $x_{\pi_{r_t}}^{q_t}$ is the sample at iteration $t = q_t N + r_t$. Let:*

$$\epsilon_t = m_t - \nabla F(w_t)$$

*Then:*

$$\mathbb{E}[\|\epsilon_t\|] \leq 2G\sqrt{\alpha} + 2\eta N L + \frac{\eta L}{\alpha}$$

*Proof.* Let $i = q_i N + r_i$ for any $i \in [t]$. Then:

$$\mathbb{E}[\|\epsilon_t\|] = \mathbb{E}[\|m_t - \nabla F(w_t)\|]$$

$$\leq \mathbb{E}[\|m_t - \alpha \sum_{i=1}^{t}(1-\alpha)^{t-i}\nabla F(w_{q_i N})\|] + \mathbb{E}[\|\alpha \sum_{i=1}^{t}(1-\alpha)^{t-i}(\nabla F(w_{q_i N}) - \nabla F(w_i))\|]$$

$$+ \mathbb{E}[\|\alpha \sum_{i=1}^{t}(1-\alpha)^{t-i}\nabla F(w_i) - \nabla F(w_t)\|]$$

$$\leq \mathbb{E}[\|m_t - \alpha \sum_{i=1}^{t}(1-\alpha)^{t-i}\nabla F(w_{q_i N})\|] + \eta N L + \mathbb{E}[\|\alpha \sum_{i=1}^{t}(1-\alpha)^{t-i}\nabla F(w_i) - \nabla F(w_t)\|]$$

$$= \mathbb{E}[\|\alpha \sum_{i=1}^{t}(1-\alpha)^{t-i}\nabla f(w_i, x_{\pi_{r_i}}^{q_i}) - \alpha \sum_{i=1}^{t}(1-\alpha)^{t-i}\nabla F(w_{q_i N})\|] + \eta N L$$

$$+ \mathbb{E}[\|\alpha \sum_{i=1}^{t}(1-\alpha)^{t-i}\nabla F(w_i) - \nabla F(w_t)\|]$$

$$\leq \mathbb{E}[\|\alpha \sum_{i=1}^{t}(1-\alpha)^{t-i}(\nabla f(w_{q_i N}, x_{\pi_{r_i}}^{q_i}) - \nabla F(w_{q_i N}))\|] + \mathbb{E}[\|\alpha \sum_{i=1}^{t}(1-\alpha)^{t-i}(\nabla f(w_i, x_{\pi_{r_i}}^{q_i}) - \nabla f(w_{q_i N}, x_{\pi_{r_i}}^{q_i}))\|]$$

$$+ \eta N L + \mathbb{E}[\|\alpha \sum_{i=1}^{t}(1-\alpha)^{t-i}\nabla F(w_i) - \nabla F(w_t)\|]$$

$$\leq \sqrt{\mathbb{E}[\|\alpha \sum_{i=1}^{t}(1-\alpha)^{t-i}(\nabla f(w_{q_i N}, x_{\pi_{r_i}}^{q_i}) - \nabla F(w_{q_i N}))\|^2]} + 2\eta N L$$

$$+ \mathbb{E}[\|\alpha \sum_{i=1}^{t}(1-\alpha)^{t-i}\nabla F(w_i) - \nabla F(w_t)\|] \qquad (5)$$

First let us bound the last term in Eq.5. Denote $g_t = (1-\alpha)g_{t-1} + \alpha \nabla F(w_t)$. Then, if we let $g_1 = \nabla F(w_1)$, $g_t = \alpha \sum_{i=1}^{t}(1-\alpha)^{t-i}\nabla F(w_i)$. Let $r_t = \mathbb{E}[\|\alpha \sum_{i=1}^{t}(1-\alpha)^{t-i}\nabla F(w_i) - \nabla F(w_t)\|] = \mathbb{E}[\|g_t - \nabla F(w_t)\|]$, we have:

$$r_t = \mathbb{E}\left[\|(1-\alpha)g_{t-1} + \alpha \nabla F(w_t) - \nabla F(w_t)\|\right]$$
$$= \mathbb{E}\left[\|(1-\alpha)(g_{t-1} - \nabla F(w_t))\|\right]$$
$$= \mathbb{E}\left[\|(1-\alpha)(g_{t-1} - \nabla F(w_{t-1})) + (1-\alpha)(\nabla F(w_{t-1}) - \nabla F(w_t))\|\right]$$

Unroll the recursive expression:

$$= \mathbb{E}\left[\|\sum_{i=1}^{t}(1-\alpha)^{t-i}((\nabla F(w_{i-1}) - \nabla F(w_i)))\|\right]$$
$$\leq \frac{\eta L}{\alpha}$$

Now let us bound $\mathbb{E}[\|\alpha \sum_{i=1}^{t}(1-\alpha)^{t-i}(\nabla f(w_{q_i N}, x_{\pi_{r_i}}^{q_i}) - \nabla F(w_{q_i N}))\|^2]$. For any iteration $i$ let $A_{\pi_{r_i}^{q_i}} = \nabla f(w_{q_i N}, x_{\pi_{r_i}}^{q_i}) - \nabla F(w_{q_i N})$ and $c_i = (1-\alpha)^{t-i}$. Thus:

$$\mathbb{E}[\|\alpha \sum_{i=1}^{t}(1-\alpha)^{t-i}(\nabla f(w_{q_i N}, x_{\pi_{r_i}}^{q_i}) - \nabla F(w_{q_i N}))\|^2] = \alpha^2 \mathbb{E}\left[\|\sum_{i=1}^{t} c_i A_{\pi_{r_i}^{q_i}}\|^2\right]$$

If we expand the equation above, we will have some cross terms as well as some squared norm terms. First, let us examine the cross terms for iteration $i < j$ where $q_i = q_j = q$. Then:

$$\mathbb{E}\left[c_i c_j \langle A_{\pi_{r_i}^q}, A_{\pi_{r_j}^q}\rangle\right] = \sum_{k_i=1}^{N} c_i c_j \mathop{\mathbb{E}}_{\pi_{r_j}^q}\left[\langle A_{\pi_{r_i}^q}, A_{\pi_{r_j}^q}\rangle | \pi_{r_i}^q = k_i\right] P[\pi_{r_i}^q = k_i]$$

$$= \sum_{k_i=1}^{N} c_i c_j P[\pi_{r_i}^q = k_i]\left\langle A_{k_i}, \mathop{\mathbb{E}}_{\pi_{r_j}^q}\left[A_{\pi_{r_j}^q} | \pi_{r_i}^q = k_i\right]\right\rangle$$

$$= \sum_{k_i=1}^{N} c_i c_j P[\pi_{r_i}^q = k_i]\left\langle A_{k_i}, \frac{\sum_{k_j \neq k_i} A_{k_j}}{N-1}\right\rangle$$

Notice that $\sum_{k_j} A_{k_j} = \sum_{k_j=1}^{N} \nabla f(w_{qN}, x_{\pi_{k_j}}^q) - \nabla F(w_{qN}) = 0$ (since the iterate at the beginning of the epoch is independent of the data that are samples *in* that particular epoch). Thus $\sum_{k_j \neq k_i} A_{k_j} = -A_{k_i}$. Then:

$$\sum_{k_i=1}^{N} c_i c_j P[\pi_{r_i}^q = k_i]\left\langle A_{k_i}, \frac{\sum_{k_j \neq k_i} A_{k_j}}{N-1}\right\rangle = \sum_{k_i=1}^{N} c_i c_j P[\pi_{r_i}^q = k_i]\left\langle A_{k_i}, \frac{-A_{k_i}}{N-1}\right\rangle$$

$$= \sum_{k_i=1}^{N} P[\pi_{r_i}^q = k_i]\frac{-c_i c_j}{N-1}\|A_{k_i}\|^2$$

$$\leq 0$$

Now let us analyze the cross terms for $i < j$ where $q_i < q_j$.

$$\mathbb{E}\left[c_i c_j \langle A_{\pi_{r_i}^{q_i}}, A_{\pi_{r_j}^{q_j}}\rangle\right] = \sum_{k_i=1}^{N} c_i c_j \mathop{\mathbb{E}}_{\pi^{q_j}}\left[\langle A_{\pi_{r_i}^{q_i}}, A_{\pi_{r_j}^{q_j}}\rangle | \pi_{r_i}^{q_i} = k_i\right] P[\pi_{r_i}^q = k_i]$$

$$= \sum_{k_i=1}^{N} c_i c_j P[\pi_{r_i}^q = k_i]\left\langle A_{\pi_{r_i}^{q_i}}, \mathop{\mathbb{E}}_{\pi^{q_j}}\left[A_{\pi_{r_j}^{q_j}} | \pi_{r_i}^{q_i} = k_i\right]\right\rangle$$

$$= \sum_{k_i=1}^{N} c_i c_j P[\pi_{r_i}^q = k_i]\left\langle A_{\pi_{r_i}^{q_i}}, \frac{\sum_{k_j=1}^{N} A_{k_j}}{N}\right\rangle$$

$$= 0$$

Thus, the cross terms $\mathbb{E}\left[c_i c_j \langle A_{\pi_{r_i}^{q_i}}, A_{\pi_{r_j}^{q_j}}\rangle\right] \leq 0$ for every $i < j$. Either way,

$$\alpha^2 \mathbb{E}\left[\|\sum_{i=1}^{t} c_i A_{\pi_{r_i}^{q_i}}\|^2\right] \leq \alpha^2 \mathbb{E}\left[\sum_{i=1}^{t} c_i^2 \|A_{\pi_{r_i}^{q_i}}\|^2\right]$$

$$\leq 4G^2\alpha^2 \sum_{i=1}^{t}(1-\alpha)^{2(t-i)}$$

$$\leq 4\alpha G^2$$

Plugging this back to eq. 5

$$\mathbb{E}[\|\epsilon_t\|] \leq 2G\sqrt{\alpha} + 2\eta N L + \frac{\eta L}{\alpha}$$

$\square$

**Lemma 12.** *Let $V = (\min(R, \lfloor \log_2(N) \rfloor) + 1)\frac{T}{N} + \sum_{j=\lfloor \log_2(N)+1 \rfloor}^{R} \lfloor \frac{T}{2^j} \rfloor$ as in Algorithm 1. Suppose $\alpha \geq \frac{1}{N}$. Then,*

$$\mathbb{E}\left[\|\hat{m}_t - m_t\|\right] \leq 4\alpha G\sigma\sqrt{dV \log_2 T}$$

*Proof.* We have:

$$\hat{m}_t = m_t + \text{NOISE}_t$$

where $\text{NOISE}_t = \sum_{[y,z]\in\text{COMPOSE}(1,t)} \zeta_{[y,z]}$. Since there are at most $\log_2 T$ intervals in COMPOSE$(1, t)$, $\text{NOISE}_t$ is a Gaussian random vector with variance:

$$\text{VAR} \leq 16\alpha^2 G^2 \sigma^2 V \log_2 T \tag{6}$$

Then:

$$\mathbb{E}\left[\|\hat{m}_t - m_t\|\right] \leq \sqrt{d}\sqrt{\text{VAR}}$$
$$= 4\alpha G\sigma\sqrt{dV \log_2 T}$$

$\square$

**Theorem 6.** *(Utility guarantee) Assuming $f(w, x)$ is G-Lipschitz, L-smooth for all $w \in \mathbb{R}^d$, $x \in \mathcal{X}$, and $F(w_1)$ is bounded by $R$. Then Algorithm 1 with $\eta = \frac{1}{\sqrt{NT}}$, $\alpha = \frac{\epsilon N}{T \log_2 T \sqrt{d \log(1/\delta)}}$, $T = \frac{\epsilon N^2}{\log_2 T \sqrt{d \log(1/\delta)}}$, $\epsilon \leq \frac{T \log_2 T \sqrt{d \log(1/\delta)}}{N}$, and $\hat{w}$ that is pick uniformly random from $w_1, \ldots, w_T$ guarantees:*

$$\mathbb{E}\left[\|\nabla F(\hat{w})\|\right] \leq \frac{(\frac{3}{2}R + 6L\sqrt{\log_2 T \log(1/\delta)} + 12G\sqrt{\log_2 T}\log(1/\delta)^{1/4})d^{1/4}}{\sqrt{\epsilon N}}$$
$$+ \frac{3L(d\log(1/\delta))^{1/4}\sqrt{\log_2 T}}{4\sqrt{\epsilon}N^{3/2}} + \frac{6G}{\sqrt{N}} + \frac{6G\log_2 T\sqrt{d\log(1/\delta)}}{\epsilon N}$$
$$\leq \tilde{O}\left(\frac{d^{1/4}}{\sqrt{\epsilon N}} + \frac{1}{\sqrt{N}}\right)$$

*Proof.* From Lemma 5, we have:

$$\mathbb{E}\left[\|\nabla F(\hat{w})\|\right] \leq \frac{3\mathbb{E}\left[(F(w_1) - F(w_{T+1}))\right]}{2\eta T} + \frac{3L\eta}{4} + \frac{3}{T}\sum_{t=1}^{T}\mathbb{E}[\|\hat{\epsilon}_t\|]$$
$$= \frac{3\mathbb{E}\left[(F(w_1) - F(w_{T+1}))\right]}{2\eta T} + \frac{3L\eta}{4} + \frac{3}{T}\sum_{t=1}^{T}\mathbb{E}[\|\hat{m}_t - \nabla F(w_t)\|]$$
$$= \frac{3\mathbb{E}\left[(F(w_1) - F(w_{T+1}))\right]}{2\eta T} + \frac{3L\eta}{4} + \frac{3}{T}\sum_{t=1}^{T}\mathbb{E}[\|\hat{m}_t - m_t + m_t - \nabla F(w_t)\|]$$
$$\leq \frac{3\mathbb{E}\left[(F(w_1) - F(w_{T+1}))\right]}{2\eta T} + \frac{3L\eta}{4} + \frac{3}{T}\sum_{t=1}^{T}\mathbb{E}\left[\|\hat{m}_t - m_t\|\right] + \mathbb{E}[\|m_t - \nabla F(w_t)\|]$$
$$\leq \frac{3\mathbb{E}\left[(F(w_1) - F(w_{T+1}))\right]}{2\eta T} + \frac{3L\eta}{4} + \frac{3}{T}\sum_{t=1}^{T}\mathbb{E}\left[\|\hat{m}_t - m_t\|\right] + \mathbb{E}[\|\epsilon_t\|]$$

Applying Lemma 11 and using Theorem 12 with $V = (\min(R, \lfloor \log_2(N) \rfloor) + 1)\frac{T}{N} + \sum_{j=\lfloor \log_2(N)+1 \rfloor}^{R} \lfloor \frac{T}{2^j} \rfloor \leq 4\log_2 T\frac{T}{N}, \sigma = \frac{2\sqrt{\log(1/\delta)}}{\epsilon}$:

$$\mathbb{E}\left[\|\nabla F(\hat{w})\|\right] \leq \frac{3R}{2\eta T} + \frac{3\eta L}{4} + 6G\sqrt{\alpha} + 6\eta NL + \frac{3\eta L}{\alpha} + \frac{12\alpha G\log_2 T\sqrt{dT\log(1/\delta)}}{\epsilon\sqrt{N}}$$
$$\leq \frac{3R}{2\eta T} + 6G\sqrt{\alpha} + 6\eta NL + \frac{6\eta L}{\alpha} + \frac{12\alpha G\log_2 T\sqrt{dT\log(1/\delta)}}{\epsilon\sqrt{N}}$$

Let $\eta = \frac{1}{\sqrt{NT}}$:

$$\mathbb{E}\left[\|\nabla F(\hat{w})\|\right] \leq \frac{(\frac{3}{2}R + 6L)\sqrt{N}}{\sqrt{T}} + \frac{6L}{\alpha\sqrt{NT}} + 6G\sqrt{\alpha} + \frac{6G}{\alpha T} + \frac{12\alpha G \log_2 T \sqrt{dT \log(1/\delta)}}{\epsilon\sqrt{N}}$$

Set $\alpha = \frac{\epsilon N}{T \log_2 T \sqrt{d \log(1/\delta)}}$:

$$\mathbb{E}\left[\|\nabla F(\hat{w})\|\right] \leq \frac{(\frac{3}{2}R + 6L + 12G)\sqrt{N}}{\sqrt{T}} + \frac{6L\sqrt{T}\log_2 T \sqrt{d\log(1/\delta)}}{\epsilon N \sqrt{N}} + \frac{6G\sqrt{\epsilon N}}{(d\log(1/\delta))^{1/4}\sqrt{T\log_2 T}}$$

$$+ \frac{6G\log_2 T \sqrt{d \, \log(1/\delta)}}{\epsilon N}$$

Since $\alpha \geq \frac{1}{N}$, then the largest $T \leq \frac{\epsilon N^2}{\log_2 T \sqrt{d\log(1/\delta)}}$:

$$\mathbb{E}\left[\|\nabla F(\hat{w})\|\right] \leq \frac{(\frac{3}{2}R + 6L\sqrt{\log_2 T \log(1/\delta)} + 12G\sqrt{\log_2 T}\log(1/\delta)^{1/4})d^{1/4}}{\sqrt{\epsilon N}} + \frac{3L(d\log(1/\delta))^{1/4}\sqrt{\log_2 T}}{4\sqrt{\epsilon}N^{3/2}} + \frac{6G}{\sqrt{N}}$$

$$+ \frac{6G\log_2 T \sqrt{d\log(1/\delta)}}{\epsilon N}$$

$$\leq \tilde{O}\left(\frac{d^{1/4}}{\sqrt{\epsilon N}} + \frac{1}{\sqrt{N}}\right)$$

$\square$

# E  Proof of section 4

## E.1  Privacy

**Theorem 8.** *(Privacy guarantee) Suppose $\alpha \geq 1/N$ and $f$ is G-Lipschitz and L-smooth. Then Algorithm 2 is $(z, 3z/2\sigma^2)$ Renyi differentially private for all $z$. With $\sigma \geq \frac{4\sqrt{\log(1/\delta)}}{\epsilon}$, Algorithm 2 is $(\epsilon, \delta)$-DP*

*Proof.* To show the privacy guarantee, we will show that the releasing all of the $\hat{F}^G_{[y,z]}$, $\hat{F}^\Delta_{[y,z]}$, $\hat{F}^r_{[y,z]}$ is private. To see this, observe that the intervals $[y, z]$ correspond to nodes of a binary tree with at least $T$ leaves: $[y, z]$ is the node whose descendents are the leaves $y, \ldots, z$, so that we are essentially analyzing a standard tree-based aggregation mechanism.

To start, let us re-define the queries:

$$F^G_{[a,b]}(x_1, \ldots, x_N) = \sum_{t=a}^{b}(1-\alpha)^{b-t}\nabla f(w_{q_t N}, x_{\pi_{r_t}^{q_t}})$$

$$F^\Delta_{[a,b]}(x_1, \ldots, x_N) = \sum_{t=a}^{b}(1-\alpha)^{b-t}(\nabla f(w_{q_t N}, x_{\pi_{r_t}^{q_t}}) - \nabla f(w_{(q_t-1)N}, x_{\pi_{r_t}^{q_t}}))$$

$$F^r_{[a,b]}(x_1, \ldots, x_N) = \sum_{t=a}^{b}(1-\alpha)^{b-t}(\nabla f(w_t, x_{\pi_{r_t}^{q_t}}) - \nabla f(w_{q_t N}, x_{\pi_{r_t}^{q_t}}))$$

First, we will compute the sensitivity of $F^G_{[a,b]}$ by the exact same analysis as in Theorem 4, we have that $F^G_{[a,b]}$ has sensitivity

$$2G \sum_{j=0}^{\lceil |b-a|/N\rceil - 1}(1-\alpha)^{jN} \leq 4G$$

where the last inequality uses Proposition 13 in conjunction with the assumption $\alpha \geq \frac{1}{N}$.

Then, for the sensitivity of $F_{[a,b]}^{\Delta}$, observe that

$$\|\nabla f(w_{q_t N}, x_{\pi_{r_t}^{q_t}}) - \nabla f(w_{q_{t'} N}, x_{\pi_{r_t}^{q_t}})\| \leq \eta N L$$

Thus, by essentially the same argument used to bound the sensitivity of $F_{[a,b]}^{G}$, $F_{[a,b]}^{\Delta}$ has sensitivity at most $2\eta N L$.

Finally, for the sensitivity of $F_{[a,b]}^{r}$:

$$\|\nabla f(w_t, x_{\pi_{r_t}^{q_t}}) - \nabla f(w_{q_t N}, x_{\pi_{r_t}^{q_t}})\| \leq \eta N L$$

Thus, $F_{[a,b]}^{r}$ also has sensitivity $2\eta N L$.

Next, observe that for any index $i \in [1, N]$, $x_i$ can influence at most $1 + \lfloor \log_2(2N) \rfloor \leq 3 \log_2(N)$ intervals $[a, b]$ corresponding to nodes in the tree such that $b \leq 2N$. Thus, by adding Gaussian noise with standard deviation $\sqrt{V_{\leq 2N}} \sigma \delta_G$ to $F_{[a,b]}^{G}$ and $\sqrt{V_{\leq 2N}} \sigma \delta_\Delta$ to $F_{[a,b]}^{\Delta}$ for any $[a, b] \subset [1, 2N]$ yields a set of estimates that are $(z, z/2\sigma^2)$ Renyi-differentially private for all $z$.

Now we turn to intervals that are not subsets of $[1, 2N]$. For these, notice that again by same analysis used to prove Theorem 4, we have that the number of nodes any index $i$ can influence is at most:

$$(\lfloor \log_2(N) \rfloor + 1) \frac{T}{N} + \sum_{j=\lfloor \log_2(N)+1 \rfloor}^{\lceil \log_2(T) \rceil} \lfloor \frac{T}{2^j} \rfloor \leq 4 \frac{T}{N} \log_2(N)$$

Thus, by adding Gaussian noise with standard deviation $\sqrt{V_{>2N}} \sigma \delta_G$ or $\sqrt{V_{>2N}} \sigma \delta_\Delta$ to $F_{[a,b]}^{G}$, or $F_{[a,b]}^{\Delta}$, we obtain $(z, z/2\sigma^2)$ Renyi-differential privacy. Finally, by adding noise with standard deviation $\sqrt{V_{>2N}} \sigma \delta_r$ to $F_{[a,b]}^{r}$, we also obtain $(z, z/2\sigma^2)$−RDP. Therefore, overall the mechanism is $(z, 3z/2\sigma^2)$−RDP.

For the $(\epsilon, \delta)$-DP guarantee, notice that $(z, 3z/2\sigma^2)$-RDP implies $(\epsilon, \delta)$-DP with $\epsilon = 3/2\sigma^2 + 2\sqrt{3 \log(1/\delta)/2\sigma^2}$ for all $\delta$. Thus, any when $\delta \geq \exp(-\epsilon)$, any $\sigma$ satisfying

$$\sigma \geq \frac{4\sqrt{\log(1/\delta)}}{\epsilon}$$

will suffice to achieve the desired privacy.

$\square$

## E.2   Utility

First, let us prove a bound on $\mathbb{E}[\|\hat{m}_t - m_t\|]$ by bounding the variance of all of the added noises.

**Lemma 7.** *Suppose $\alpha \geq \frac{1}{N}$ and $\gamma \leq \frac{1}{2}$, using the update of Algorithm 2, we have:*

$$\mathbb{E}[\|\hat{m}_t - m_t\|] \leq \frac{192 G \alpha \log_2 T \sqrt{d \log(1/\delta)}}{\epsilon} + \frac{128(G + 2L)\alpha \sqrt{\eta dT \log(1/\delta)} \log_2 T}{\epsilon}$$
$$+ \frac{16 \eta L \alpha \log_2 T \sqrt{3dTN \log(1/\delta)}}{\epsilon}$$

*Proof.* First, observe that since $\alpha \geq 1/N$, by Proposition 13, we have:

$$\sum_{i=0}^{\infty} (1 - \alpha)^{iN} \leq \frac{1}{1 - \exp(-1)} \leq 2 \tag{7}$$

Now, we define some notation. Let For any iteration t, let $t = q_t N + r_t$. For any interval $[a, b]$, set

$$F^G_{[a,b]}(x_1, \ldots, x_N) = \sum_{t=a}^{b} (1-\alpha)^{b-t} \nabla f(w_{q_t N}, x_{\pi^{q_t}_{r_t}})$$

$$F^\Delta_{[a,b]}(x_1, \ldots, x_N) = \sum_{t=a}^{b} (1-\alpha)^{b-t} (\nabla f(w_{q_t N}, x_{\pi^{q_t}_{r_t}}) - \nabla f(w_{(q_t-1)N}, x_{\pi^{q_t}_{r_t}}))$$

$$F^r_{[a,b]}(x_1, \ldots, x_N) = \sum_{t=a}^{b} (1-\alpha)^{b-t} (\nabla f(w_t, x_{\pi^{q_t}_{r_t}}) - \nabla f(w_{q_t N}, x_{\pi^{q_t}_{r_t}}))$$

We will use the notation $F^G_{[a,b]}$ to indicate $F^G_{[a,b]}(x_1, \ldots, x_N)$ for brevity. Then:

$$G_{[a,b]} = \sum_{[y,z] \in \text{COMPOSE}(a,b)} (1-\alpha)^{b-z} F^G_{[y,z]}$$

$$\Delta_{[a,b]} = \sum_{[y,z] \in \text{COMPOSE}(a,b)} (1-\alpha)^{b-z} F^\Delta_{[y,z]}$$

$$r_{[a,b]} = \sum_{[y,z] \in \text{COMPOSE}(a,b)} (1-\alpha)^{b-z} F^r_{[y,z]}$$

Observe that with this definition, if $m_t$ is the momentum value defined recursively as $m_{t+1} = (1-\alpha)m_t + \alpha\nabla f(w_t, x_{\pi^{q_t}_{r_t}})$, we have:

$$m_t = \alpha \sum_{i=1}^{t} (1-\alpha)^{t-i} \nabla f(w_i, x_{\pi^{q_i}_{r_i}})$$

$$= \alpha \sum_{i=1}^{t} (1-\alpha)^{t-i} \nabla f(w_{q_i N}, x_{\pi^{q_i}_{r_i}}) + \alpha \sum_{i=1}^{t} (1-\alpha)^{t-i} (\nabla f(w_i, x_{\pi^{q_i}_{r_i}}) - \nabla f(w_{q_i N}, x_{\pi^{q_i}_{r_i}}))$$

$$= \alpha G_{[1,t]} + \alpha r_{[1,t]}$$

From Lemma 17, we know that:

$$\alpha G_{[1,t]} = (1-\alpha)^{t-r} \alpha G_{[1,r]} + \sum_{i=0}^{q-1} (1-\alpha)^{Ni} (1-\gamma)^{q-(i+1)} \alpha G_{[r+1,r+N]}$$

$$+ \sum_{i=1}^{q-1} \left( \sum_{j=0}^{i-1} (1-\gamma)^j (1-\alpha)^{(i-1-j)N} \right) \alpha \left( (1-\gamma)\Delta_{[t-iN+1, t-(i-1)N]} + \gamma G_{[t-iN+1, t-(i-1)N]} \right)$$

$$\tag{8}$$

Now we can compute the accuracy of the estimate $\alpha\hat{G}_{[1,t]}$ using $\delta_\Delta$, $\delta_G$, $\delta_r$, $V_{\leq 2N} = 3\log_2 N$, $V_{>2N} = 4\log_2 N \frac{T}{N}$. Let us analyze the noise added to $\alpha\hat{G}_{[1,t]}$ term by term.

**First term, $(1-\alpha)^{T-r}\alpha G_{[1,r]}$:**

$$(1-\alpha)^{T-r}\alpha\hat{G}_{[1,r]} = (1-\alpha)^{T-r}\alpha \sum_{[y,z] \in \text{COMPOSE}(1,r)} (1-\alpha)^{r-z} F^G_{[y,z]} + (1-\alpha)^{T-r}\alpha \sum_{[y,z] \in \text{COMPOSE}(1,r)} (1-\alpha)^{r-z} \zeta^G_{[y,z]}$$

$$= (1-\alpha)^{T-1}\alpha G_{[1:r]} + (1-\alpha)^{T-r}\alpha \sum_{[y,z] \in \text{COMPOSE}(1,r)} (1-\alpha)^{r-z} \zeta^G_{[y,z]}$$

where $\zeta^G_{[y,z]} \sim N(0, \delta^2_G \sigma^2 V)$ with $V_{\leq 2N} = 3\log_2 N$. Now, by Proposition 18, there are at most $2(1 + \log_2 r) \leq 4\log_2(N)$ intervals in $\text{COMPOSE}(1,r)$. Thus, the variance of the noise added to the first term is:

$$\text{VAR}_1 \leq 2(1-\alpha)^{T-r}(1 + \log_2(r))\alpha^2\delta^2_G\sigma^2 V$$

$$\leq 12(1-\alpha)^{T-r}\alpha^2\delta^2_G\sigma^2 \log_2^2 N$$

$$\leq 12\alpha^2\delta^2_G\sigma^2 \log_2^2 N$$

**Second term, $\sum_{i=0}^{q-1}(1-\alpha)^{Ni}(1-\gamma)^{q-(i+1)}\alpha G_{[r+1,r+N]}$:**

$$\sum_{i=0}^{q-1}(1-\alpha)^{Ni}(1-\gamma)^{q-(i+1)}\alpha G_{[r+1,r+N]} = \sum_{i=0}^{q-1}(1-\alpha)^{Ni}(1-\gamma)^{q-(i+1)}\alpha \sum_{[y,z]\in\text{COMPOSE}(r+1,r+N)} \left(F_{[y,z]}^{G} + \zeta_{[y,z]}^{G}\right)$$

Since there are at most $4\log_2 N$ terms in $\text{COMPOSE}(1,N)$, the variance of the noise added to the second term is:

$$\text{VAR}_2 \le 4\sum_{i=0}^{q-1}(1-\alpha)^{2Ni}(1-\gamma)^{2q-2(i+1)}\log_2 N\alpha^2\delta_G^2\sigma^2 V_{\le 2N}$$

$$\le \frac{4\log_2 N\alpha^2\delta_G^2\sigma^2 V_{\le 2N}}{(1-\exp(-1))^2}$$

$$= \frac{12\alpha^2\delta_G^2\sigma^2\log_2^2 N}{(1-\exp(-1))^2}$$

where the second inequality comes from Proposition 14.

**Third term, $\alpha\sum_{i=1}^{q-1}\left(\sum_{j=0}^{i-1}(1-\gamma)^j(1-\alpha)^{(i-1-j)N}\right)\gamma G_{[T-iN+1,T-(i-1)N]}$:**

$$\sum_{i=1}^{q-1}\left(\sum_{j=0}^{i-1}(1-\gamma)^j(1-\alpha)^{(i-1-j)N}\right)\alpha\gamma\hat{G}_{[T-iN+1,T-(i-1)N]}$$

$$= \sum_{i=1}^{q-1}\left(\sum_{j=0}^{i-1}(1-\gamma)^j(1-\alpha)^{(i-1-j)N}\right)\alpha\gamma \sum_{[y,z]\in\text{COMPOSE}(T-iN+1,T-(i-1)N)} (1-\alpha)^{T-(i-1)N-z}(F_{[y,z]}^{G} + \zeta_{[y,z]}^{G})$$

where $\zeta_{[y,z]}^{G} \sim N(0,\delta_G^2\sigma^2 V_{>2N})$ where $V_{>2N} = 4\frac{T}{N}\log_2 N$. There are at still at most $4\log_2(N)$ terms in $\text{COMPOSE}(T-iN+1,T-(i-1)N)$, so Using Corollary 15, the variance of the noise added to the third term is:

$$\text{VAR}_3 \le 4\sum_{i=1}^{q-1}\frac{(1-\gamma)^{2(i-1)}}{(1-\exp(-1))^2}\alpha^2\gamma^2\log_2 N\delta_G^2\sigma^2 V_{>2N}$$

$$\le \frac{16\gamma\alpha^2\log_2^2 N\delta_G^2\sigma^2 T}{(1-\exp(-1))^2 N}$$

$$\le \frac{16\gamma\alpha^2\delta_G^2\sigma^2 T\log_2^2 N}{(1-\exp(-1))^2 N}$$

**Fourth Term, $\sum_{i=1}^{q-1}\left(\sum_{j=0}^{i-1}(1-\gamma)^j(1-\alpha)^{(i-1-j)N}\right)\alpha(1-\gamma)\Delta_{[T-iN+1,T-(i-1)N]}$:**

Similar to the third term, we have:

$$\sum_{i=1}^{q-1}\left(\sum_{j=0}^{i-1}(1-\gamma)^j(1-\alpha)^{(i-1-j)N}\right)\alpha(1-\gamma)\hat{\Delta}_{[T-iN+1,T-(i-1)N]}$$

$$= \sum_{i=1}^{q-1}\left(\sum_{j=0}^{i-1}(1-\gamma)^j(1-\alpha)^{(i-1-j)N}\right)\alpha(1-\gamma) \sum_{[y,z]\in\text{COMPOSE}(T-iN+1,T-(i-1)N)} (1-\alpha)^{T-(i-1)N-z}(\Delta_{[y,z]} + \zeta_{[y,z]}^{\Delta})$$

Thus the variance of the noise added to the fourth term is:

$$\text{VAR}_4 \le 4\sum_{i=1}^{q-1}\alpha^2\frac{(1-\gamma)^{2i}}{(1-\exp(-1))^2}\log_2 N\delta_\Delta^2\sigma^2 V_{>2N}$$

$$\le \frac{16\alpha^2\log_2^2 N\delta_\Delta^2\sigma^2 T}{\gamma N(1-\exp(-1))^2}$$

$$\le \frac{16\alpha^2\delta_\Delta^2\sigma^2 T\log_2^2 N}{\gamma N(1-\exp(-1))^2}$$

Now let us analyze $\alpha r_{[1,t]}$ to see how much noise we need to add to make it private. We have:

$$\alpha\hat{r}_{[1,t]} = \alpha r_{[1,t]} + \alpha \sum_{[y,z]\in \mathrm{COMPOSE}(1,t)} \zeta^r_{[y,z]}$$

where $\zeta^r_{[y,z]} \sim N(0, \delta^2_r \sigma^2 V_{>2N})$. There are at most $3\log_2 T$ in $\mathrm{COMPOSE}(1,t)$, thus the variance of the noise added is:

$$\mathrm{VAR}_5 = 3\log_2 T\alpha^2\delta^2_r\sigma^2 V_{>2N}$$

Plug in $V_{>2N} = 4\frac{T}{N}\log_2 N$, $\delta^r = 2\eta NL$

$$\leq \frac{48T\alpha^2\eta^2 N^2 L^2\sigma^2 \log^2_2 T}{N}$$

Now combining $\mathrm{VAR}_1$, $\mathrm{VAR}_2$, $\mathrm{VAR}_3$, $\mathrm{VAR}_4$, $\mathrm{VAR}_5$ we have the total variance of the Gaussian noise added to $\hat{m}_t$ is:

$$\mathrm{VAR} \leq 12\alpha^2\delta^2_G\sigma^2\log^2_2 N + \frac{12\alpha^2\delta^2_G\sigma^2\log^2_2 N}{(1-\exp(-1))^2} + \frac{16\gamma\alpha^2\delta^2_G\sigma^2 T\log^2_2 N}{(1-\exp(-1))^2 N} + \frac{16\alpha^2\delta^2_\Delta\sigma^2 T\log^2_2 N}{(1-\exp(-1))^2\gamma N}$$

$$+ \frac{48T\alpha^2\eta^2 N^2 L^2\sigma^2\log^2_2 T}{N}$$

$$= 256G^2\alpha^2\sigma^2\log^2_2 N + \frac{256G^2\alpha^2\sigma^2\log^2_2 N}{(1-\exp(-1))^2} + \frac{256G^2\gamma\alpha^2\sigma^2 T\log^2_2 N}{(1-\exp(-1))^2 N} + \frac{1024\eta^2 N^2 L^2\alpha^2\sigma^2 T\log^2_2 N}{(1-\exp(-1))^2\gamma N}$$

$$+ \frac{48T\alpha^2\eta^2 N^2 L^2\sigma^2\log^2_2 T}{N}$$

Since the added noise is a Gaussian vector:

$$\mathbb{E}[\|\hat{m}_t - m_t\|]$$
$$\leq \sqrt{d}\sqrt{\mathrm{VAR}}$$
$$\leq \sqrt{d}\times\left(16G\alpha\sigma\log_2 N + \frac{16G\alpha\sigma\log_2 N}{1-\exp(-1)} + \frac{16G\alpha\sqrt{\gamma}\sigma\sqrt{T}\log_2 N}{(1-\exp(-1))\sqrt{N}} + \frac{32\eta NL\alpha\sigma\sqrt{T}\log_2 N}{(1-\exp(-1))\sqrt{\gamma N}} + \frac{4\eta NL\alpha\sigma\log_2 T\sqrt{3T}}{\sqrt{N}}\right)$$
$$\leq \sqrt{d}\times\left(48G\alpha\sigma\log_2 T + \frac{32G\alpha\sqrt{\gamma}\sigma\sqrt{T}\log_2 T}{\sqrt{N}} + \frac{64\eta NL\alpha\sigma\sqrt{T}\log_2 T}{\sqrt{\gamma N}} + \frac{4\eta NL\alpha\sigma\log_2 T\sqrt{3T}}{\sqrt{N}}\right)$$

Set $\gamma = \eta N$:

$$\mathbb{E}[\|\hat{m}_t - m_t\|] \leq \sqrt{d}\times\left(48G\alpha\sigma\log_2 T + 32G\alpha\sigma\sqrt{\eta T}\log_2 T + 64L\alpha\sigma\sqrt{\eta T}\log_2 T + \frac{4\eta NL\alpha\sigma\log_2 T\sqrt{3T}}{\sqrt{N}}\right)$$

Setting $\sigma = \frac{4\sqrt{\log(1/\delta)}}{\epsilon}$:

$$\mathbb{E}[\|\hat{m}_t - m_t\|] \leq \frac{192G\alpha\log_2 T\sqrt{d\log(1/\delta)}}{\epsilon} + \frac{128(G+2L)\alpha\sqrt{\eta d T\log(1/\delta)}\log_2 T}{\epsilon} + \frac{16\eta L\alpha\log_2 T\sqrt{3dTN\log(1/\delta)}}{\epsilon}$$

$$\square$$

**Theorem 9.** *(Utility guarantee) Assuming $f(w,x)$ is G-Lipschitz, L-smooth for all $w \in \mathbb{R}^d$, $x \in \mathcal{X}$, and $F(w_1)$ is bounded by R. Then Algorithm 2 with $\gamma = \eta N$, $\eta = \frac{1}{\sqrt{NT}}$, $\alpha = \frac{N^{3/4}\epsilon}{T^{3/4}\sqrt{d}}$, $T = \frac{N^{7/3}\epsilon^{4/3}}{d^{2/3}}$, $\epsilon \leq \frac{T^{3/4}\sqrt{d}}{N^{3/4}}$, and $\hat{w}$ that is picked uniformly at random from $w_1, ..., w_T$ guarantees:*

$$\mathbb{E}[\|\nabla F(\hat{w})\|] \leq \frac{(\frac{3}{2}R + 24K(G+2L) + 12L)d^{1/3}}{(\epsilon N)^{2/3}} + \frac{6G}{\sqrt{N}} + \frac{36GK\sqrt{d}}{\epsilon N} + \frac{9KL\sqrt{d}}{\epsilon N}$$

$$\leq \tilde{O}\left(\frac{d^{1/3}}{(\epsilon N)^{2/3}} + \frac{1}{\sqrt{N}}\right)$$

*where $K = 16\log_2 T\log(1/\delta)$.*

*Proof.* From Lemma 7, we have:

$$\mathbb{E}[\|\hat{m}_t - m_t\|] \leq \frac{192 G\alpha \log_2 T \sqrt{d \log(1/\delta)}}{\epsilon} + \frac{128(G + 2L)\alpha\sqrt{\eta dT \log(1/\delta)}\log_2 T}{\epsilon}$$
$$+ \frac{16\eta L\alpha \log_2 T\sqrt{3dTN \log(1/\delta)}}{\epsilon}$$

Now use Lemma 11, 5 and let $K = 16\log_2 T \log(1/\delta)$:

$$\mathbb{E}[\|\nabla F(\hat{w})\|] \leq \frac{3R}{2\eta T} + \frac{3\eta L}{4}$$
$$+ \frac{3}{T}\sum_{t=1}^{T}\left(2G\sqrt{\alpha} + 2\eta NL + \frac{\eta L}{\alpha} + \frac{12KG\alpha\sqrt{d}}{\epsilon} + \frac{8K(G+2L)\alpha\sqrt{\eta dT}}{\epsilon} + \frac{K\alpha\eta L\sqrt{3dTN}}{\epsilon}\right)$$
$$\leq \frac{3R}{2\eta T} + \frac{3\eta L}{4} + 6G\sqrt{\alpha} + 6\eta NL + \frac{3\eta L}{\alpha} + \frac{36GK\alpha\sqrt{d}}{\epsilon} + \frac{24K(G+2L)\alpha\sqrt{\eta dT}}{\epsilon} + \frac{3K\alpha\eta L\sqrt{3dTN}}{\epsilon}$$
$$\leq \frac{3R}{2\eta T} + 6G\sqrt{\alpha} + 6\eta NL + \frac{6\eta L}{\alpha} + \frac{36GK\alpha\sqrt{d}}{\epsilon} + \frac{24K(G+2L)\alpha\sqrt{\eta dT}}{\epsilon} + \frac{3K\alpha\eta L\sqrt{3dTN}}{\epsilon}$$

Set $\eta = \frac{1}{\sqrt{NT}}$:

$$\mathbb{E}[\|\nabla F(\hat{w})\|] \leq \frac{(\frac{3}{2}R + 6L)\sqrt{N}}{\sqrt{T}} + \frac{6L}{\alpha\sqrt{NT}} + 6G\sqrt{\alpha} + \frac{36GK\alpha\sqrt{d}}{\epsilon} + \frac{24K(G+2L)\alpha T^{1/4}\sqrt{d}}{\epsilon N^{1/4}} + \frac{3K\alpha L\sqrt{3d}}{\epsilon}$$
$$\leq \frac{(\frac{3}{2}R + 6L)\sqrt{N}}{\sqrt{T}} + \frac{6L}{\alpha\sqrt{NT}} + 6G\sqrt{\alpha} + \frac{36GK\alpha\sqrt{d}}{\epsilon} + \frac{24K(G+2L)\alpha T^{1/4}\sqrt{d}}{\epsilon N^{1/4}} + \frac{9K\alpha L\sqrt{d}}{\epsilon}$$

Set $\alpha = \frac{N^{3/4}\epsilon}{T^{3/4}\sqrt{d}}$:

$$\mathbb{E}[\|\nabla F(\hat{w})\|] \leq \frac{(\frac{3}{2}R + 24K(G+2L) + 6L)\sqrt{N}}{\sqrt{T}} + \frac{6LT^{1/4}\sqrt{d}}{\epsilon N^{5/4}} + \frac{6GN^{3/8}\sqrt{\epsilon}}{T^{3/8}d^{1/4}} + \frac{36GKN^{3/4}}{T^{3/4}} + \frac{9KN^{3/4}L}{T^{3/4}}$$

Because we must have $\alpha \geq \frac{1}{N}$, the largest value of $T = \frac{N^{7/3}\epsilon^{4/3}}{d^{2/3}}$:

$$\mathbb{E}[\|\nabla F(\hat{w})\|] \leq \frac{(\frac{3}{2}R + 24K(G+2L) + 12L)d^{1/3}}{(\epsilon N)^{2/3}} + \frac{6G}{\sqrt{N}} + \frac{36GK\sqrt{d}}{\epsilon N} + \frac{9KL\sqrt{d}}{\epsilon N}$$
$$\leq \tilde{O}\left(\frac{d^{1/3}}{(\epsilon N)^{2/3}} + \frac{1}{\sqrt{N}}\right)$$

$\square$

# F   Technical Lemmas

**Lemma 5.** *[Essentially Lemma 2 [Cutkosky and Mehta [2020]] Define:*
$$\hat{\epsilon}_t = \hat{m}_t - \nabla F(w_t)$$

*Suppose $w_1, ..., w_T$ is a sequence of iterates defined by $w_{t+1} = w_t - \eta\frac{\hat{m}_t}{\|\hat{m}_t\|}$ for some arbitrary sequence $\hat{m}_1, ..., \hat{m}_T$. Pick $\hat{w}$ uniformly at random from $w_1, ..., w_T$. Then:*

$$\mathbb{E}[\|\nabla F(\hat{w})\|] \leq \frac{3\,\mathbb{E}[(F(w_1) - F(w_{T+1}))]}{2\eta T} + \frac{3L\eta}{4} + \frac{3}{T}\sum_{t=1}^{T}\mathbb{E}[\|\hat{\epsilon}_t\|]$$

*Proof.* From smoothness:

$$F(w_{t+1}) \leq F(w_t) + \langle\nabla F(w_t), w_{t+1} - w_t\rangle + \frac{L}{2}\|w_{t+1} - w_t\|^2$$
$$= F(w_t) + \eta\left\langle\nabla F(w_t), \frac{\hat{m}_t}{\|\hat{m}_t\|}\right\rangle + \frac{L\eta^2}{2}$$

Let's analyze the inner-product term via some case-work: Suppose $\|\hat{\epsilon}_t\| \leq \frac{1}{2}\|\nabla F(w_t)\|$. Then $\frac{1}{2}\|\nabla F(w_t)\| \leq \|\nabla F(w_t) + \hat{\epsilon}_t\| \leq \frac{3}{2}\|\nabla F(w_t)\|$ so that:

$$-\left\langle \nabla F(w_t), \frac{\hat{m}_t}{\|\hat{m}_t\|} \right\rangle = -\left\langle \nabla F(w_t), \frac{\nabla F(w_t) + \hat{\epsilon}_t}{\|\nabla F(w_t) + \hat{\epsilon}_t\|} \right\rangle$$

$$\leq \frac{-\|\nabla F(w_t)\|^2}{\|\nabla F(w_t) + \hat{\epsilon}_t\|} + \frac{\|\nabla F(w_t)\|\|\hat{\epsilon}_t\|}{\|\nabla F(w_t) + \hat{\epsilon}_t\|}$$

$$\leq -\frac{2}{3}\|\nabla F(w_t)\| + 2\|\hat{\epsilon}_t\|$$

On the other hand, if $\|\hat{\epsilon}_t\| > \frac{1}{2}\|\nabla F(w_t)\|$, then

$$-\left\langle \nabla F(w_t), \frac{\hat{m}_t}{\|\hat{m}_t\|} \right\rangle \leq 0$$

$$\leq -\frac{2}{3}\|\nabla F(w_t)\| + \frac{2}{3}\|\nabla F(w_t)\|$$

$$\leq -\frac{2}{3}\|\nabla F(w_t)\| + \frac{4}{3}\|\hat{\epsilon}_t\|$$

So either way, we have $-\left\langle \nabla F(w_t), \frac{\hat{m}_t}{\|\hat{m}_t\|} \right\rangle \leq -\frac{2}{3}\|\nabla F(w_t)\| + 2\|\hat{\epsilon}_t\|$ Now sum over t and rearrange to get:

$$\frac{1}{T}\sum_{t=1}^{T}\|\nabla F(w_t)\| \leq \frac{3(F(w_1) - F(w_{T+1}))}{2\eta T} + \frac{3L\eta}{4} + \frac{3}{T}\sum_{t=1}^{T}\|\hat{\epsilon}_t\|$$

Take expectation of both sides:

$$\frac{1}{T}\mathbb{E}\left[\sum_{t=1}^{T}\|\nabla F(w_t)\|\right] \leq \frac{3\,\mathbb{E}\left[(F(w_1) - F(w_{T+1}))\right]}{2\eta T} + \frac{3L\eta}{4} + \frac{3}{T}\sum_{t=1}^{T}\mathbb{E}[\|\hat{\epsilon}_t\|]$$

Pick $\hat{w}$ uniformly at random from $w_1, ..., w_T$. Then:

$$\mathbb{E}\left[\|\nabla F(\hat{w})\|\right] \leq \frac{3\,\mathbb{E}\left[(F(w_1) - F(w_{T+1}))\right]}{2\eta T} + \frac{3L\eta}{4} + \frac{3}{T}\sum_{t=1}^{T}\mathbb{E}[\|\hat{\epsilon}_t\|]$$

$\square$

**Proposition 13.** *Let $\alpha \in (0, 1]$ and $N$ be an arbitrary positive integer. Then:*

$$\sum_{i=0}^{\infty}(1 - \alpha)^{Ni} \leq \frac{1}{1 - \exp(-1)}\max\left(1, \frac{1}{\alpha N}\right)$$

*Proof.*

$$\sum_{i=0}^{\infty}(1 - \alpha)^{Ni} = \sum_{i=0}^{\infty}\left((1 - \alpha)^N\right)^i$$

using the identity $1 - x \leq \exp(-x)$:

$$\leq \sum_{i=0}^{\infty}(\exp(-\alpha N))^i$$

Now, if $\alpha N \leq 1$, then we use the identity $\exp(-x) \leq 1 - x(1 - \exp(-1))$ for all $x \in [0, 1]$:

$$\sum_{i=0}^{\infty}(1 - \alpha)^{Ni} \leq \sum_{i=0}^{\infty}[1 - (1 - \exp(-1))\alpha N]^i$$

$$= \frac{1}{\alpha N(1 - \exp(-1))}$$

On the other hand, if $\alpha N > 1$, then we have:

$$\sum_{i=0}^{\infty} (1-\alpha)^{Ni} \leq \sum_{i=0}^{\infty} (\exp(-\alpha N))^i$$

$$\leq \sum_{i=0}^{\infty} \exp(-i)$$

$$\leq \frac{1}{1 - \exp(-1)}$$

Putting the two cases together yields the desired result. $\qquad \square$

**Proposition 14.** *Let* $\gamma \in (0,1]$ $\frac{1}{N} \ln\left(\frac{1}{1-\gamma}\right) \leq \alpha \leq 1$ *and let* $N$ *be an arbitrary positive integer. Then:*

$$\sum_{i=0}^{\infty} (1-\gamma)^{-i}(1-\alpha)^{Ni} \leq \frac{1}{1-\exp(-1)} \max\left(1, \frac{1}{\alpha N - \ln\left(\frac{1}{1-\gamma}\right)}\right)$$

*In particular, if* $\gamma \leq 1/2$ *and* $\frac{1+\ln(2)}{N} \leq \alpha$:

$$\sum_{i=0}^{\infty} (1-\gamma)^{-i}(1-\alpha)^{Ni} \leq \frac{1}{1-\exp(-1)}$$

*Proof.*

$$\sum_{i=0}^{\infty} (1-\gamma)^{-i}(1-\alpha)^{Ni} = \sum_{i=0}^{\infty} \left((1-\alpha)^N\right)^i$$

using the identity $1 - x \leq \exp(-x)$:

$$\leq \sum_{i=0}^{\infty} (1-\gamma)^i \left(\exp(-\alpha N)\right)^i$$

$$= \sum_{i=0}^{\infty} \left[\exp\left(-\alpha N + \ln\left(\frac{1}{1-\gamma}\right)\right)\right]^i$$

Now, as in the proof of Proposition 13, we consider two cases. First, if $0 \leq \alpha N - \ln\left(\frac{1}{1-\gamma}\right) \leq 1$, then since $\exp(-x) \leq 1 - (1 - \exp(-1))x$ for all $x \in [0,1]$, we have:

$$\sum_{i=0}^{\infty} (1-\gamma)^{-i}(1-\alpha)^{Ni} \leq \sum_{i=0}^{\infty} \left[1 - (1-\exp(-1))\left(\alpha N - \ln\left(\frac{1}{1-\gamma}\right)\right)\right]^i$$

$$\leq \frac{1}{(1-\exp(-1))\left(\alpha N - \ln\left(\frac{1}{1-\gamma}\right)\right)}$$

Alternatively, if $\alpha N - \ln\left(\frac{1}{1-\gamma}\right) \geq 1$,

$$\sum_{i=0}^{\infty} (1-\gamma)^{-i}(1-\alpha)^{Ni} \leq \sum_{i=0}^{\infty} \exp(-i)$$

$$\leq \frac{1}{1-\exp(-1)}$$

Putting the two cases together provides the first statement in the Proposition.

For the second statement, observe that since $\ln\left(\frac{1}{1-\gamma}\right)$ is increasing in $\gamma$, we have $\frac{1+\ln(2)}{N} \leq \alpha$ implies $1 + \ln\left(\frac{1}{1-\gamma}\right) \leq N\alpha$, from which the result follows. $\qquad \square$

**Corollary 15.** *Suppose $\gamma \in (0,1]$ and $1 \geq \alpha \geq \frac{1}{N} \ln \left( \frac{1}{1-\gamma} \right)$: Then:*

$$\sum_{j=0}^{i-1} (1-\gamma)^j (1-\alpha)^{(i-1-j)N} \leq \frac{(1-\gamma)^{i-1}}{1-\exp(-1)} \max \left( 1, \frac{1}{\alpha N - \ln \left( \frac{1}{1-\gamma} \right)} \right)$$

*Proof.*

$$\sum_{j=0}^{i-1} (1-\gamma)^j (1-\alpha)^{(i-1-j)N} = (1-\gamma)^{i-1} \sum_{j=0}^{i-1} (1-\gamma)^{j-(i-1)} (1-\alpha)^{(i-1-j)N}$$

$$\leq (1-\gamma)^{i-1} \sum_{k=0}^{\infty} (1-\gamma)^{-k} (1-\alpha)^{kN}$$

now apply Proposition 14:

$$\leq \frac{(1-\gamma)^{i-1}}{1-\exp(-1)} \max \left( 1, \frac{1}{\alpha N - \ln \left( \frac{1}{1-\gamma} \right)} \right)$$

$\square$

**Proposition 16.** *Let $\gamma \in (0,1)$ and $s$ be any integer. Suppose $b_R b_{R-1} \ldots b_0$ is the binary expansion of $s$, so that $s = \sum_{i=0}^{R} b_i 2^i$. Then:*

$$\sum_{i=0}^{R} b_i \gamma^{2(s \mod 2^i)} \leq 1 + \sum_{i=1}^{R} \gamma^{2^i}$$

*Proof.* Define $F(s) = \sum_{i=0}^{R} b_i \gamma^{2(s \mod 2^i)}$ where $b_i$ is the $i$th bit of the binary expansion of $s$. Then we will show first:

$$F \left( \sum_{i=0}^{R} 2^i \right) \leq 1 + \sum_{i=1}^{R} \gamma^{2^i}$$

and then for all $s < 2^{R+1}$,

$$F(s) \leq F \left( \sum_{i=0}^{R} 2^i \right)$$

which together proves the Proposition.

For the first claim, we have:

$$\sum_{i=0}^{R} 2^i \mod 2^j = 2^j - 1 \text{ for all } j \leq R+1$$

so that:

$$F \left( \sum_{i=0}^{R} 2^i \right) = \sum_{i=0}^{R} \gamma^{2(2^i-1)}$$

$$= 1 + \sum_{i=1}^{R} \gamma^{2(2^i-1)}$$

$$\leq 1 + \sum_{i=1}^{R} \gamma^{2^i}$$

Now, for the second claim, suppose that $s \neq \sum_{i=0}^{R} 2^i$. That is, there is some $j$ such that $b_j = 0$. Equivalently, we can write $s = A + B$ such that $A < 2^j$ and $B = 0 \mod 2^{j+1}$ for some $j \leq R$.

Then define $s' = A + B/2$. Let $b'_R \ldots b'_0$ be the binary expansion of $s'$. Then we have $b'_i = b_i$ for $i < j$ and $b'_i = b_{i+1}$ for $i \geq j$. Further, for $i < j$, $s \mod 2^i = A \mod 2^i = s' \mod 2^i$, and for $i \geq j$, $s \mod 2^{i+1} = A + (B \mod 2^{i+1}) \geq A + (B/2 \mod 2^i)$. Thus, we have:

$$F(s) = \sum_{i=0}^{R} b_i \gamma^{2(s \mod 2^i)} \leq \sum_{i=0}^{R} b'_i \gamma^{2(s' \mod 2^i)} = F(s')$$

By repeating this argument, we see that if $s$ has $n$ non-zero bits in the binary expansion, $F(s) \leq F(2^n - 1)$. Finally, notice that adding higher-order bits to the binary expansion can only increase $F$, so that $F(2^n - 1) \leq F(2^{R+1} - 1)$ and so we are done.

$\square$

**Lemma 17.** *Let $T = qN + r$ where $q \in \mathbb{Z}$ is the quotient and $r \in [0, N-1]$ is the remainder when dividing $T$ by $N$. Then:*

$$G_{[1,T]} = (1-\alpha)^{T-r} G_{[1,r]} + \sum_{i=0}^{q-1} (1-\alpha)^{Ni}(1-\gamma)^{q-(i+1)} G_{[r+1,r+N]}$$

$$+ \sum_{i=1}^{q-1} \left( \sum_{j=0}^{i-1} (1-\gamma)^j (1-\alpha)^{(i-1-j)N} \right) \left( (1-\gamma)\Delta_{[T-iN+1,T-(i-1)N]} + \gamma G_{[T-iN+1,T-(i-1)N]} \right)$$

*Proof.* For any iteration t, let $t = q_t N + r_t$. We re-define:

$$F^G_{[a,b]} = \sum_{i=a}^{b} (1-\alpha)^{b-i} \nabla f(w_{q_i N}, x_{\pi^{q_i}_{r_i}})$$

$$F^\Delta_{[a,b]} = \sum_{i=a}^{b} (1-\alpha)^{b-i} (\nabla f(w_{q_i N}, x^{q_i}_{r_i}) - \nabla f((w_{(q_i-1)N}, x^{q_i}_{r_i})))$$

and:

$$G_{[a,b]} = \sum_{[y,z] \in \text{COMPOSE}(a,b)} (1-\alpha)^{b-z} F^G_{[y,z]}$$

$$\Delta_{[a,b]} = \sum_{[y,z] \in \text{COMPOSE}(a,b)} (1-\alpha)^{b-z} F^\Delta_{[y,z]}$$

Interpreting $\sum_{i=a}^{b}$ as 0 whenever $b < a$, we see that statement of the Lemma is immediate for $T < N$, as $q = 0$, $r = T$ and all sums are zero. Next, a little calculation reveals the following identity:

$$G_{[1,T]} = (1-\alpha)^N G_{[1,T-N]} + G_{[T-N+1,T]} \tag{9}$$

which implies the Lemma for $q = 1$ (since $T - N = r$).

Further, we have:

$$G_{[T-N+1,T]} = (1-\gamma)\Delta_{[T-N+1,T]} + \gamma G_{[T-N+1,T]} + (1-\gamma)G_{[T-2N+1,T-N]}$$

Putting these together yields:

$$G_{[1,T]} = (1-\alpha)^N G_{[1,T-N]} + (1-\gamma)\Delta_{[T-N+1,T]} + \gamma G_{[T-N+1,T]} + (1-\gamma)G_{[T-2N+1,T-N]}$$

$$= (1-\alpha)^{2N} G_{[1,T-2N]} + (1-\alpha)^N G_{[T-2N+1,T-N]} + (1-\gamma)\Delta_{[T-N+1,T]} + \gamma G_{[t-N+1,T]}$$

$$+ (1-\gamma)G_{[T-2N+1,T-N]}$$

$$= (1-\alpha)^{2N}(G_{[1,T-2N]} + ((1-\alpha)^N + (1-\gamma))G_{[T-2N+1,T-N]} + (1-\gamma)\Delta_{[T-N+1,T]} + \gamma G_{[T-N+1,T]}$$

which is exactly the statement of the Lemma for $q = 2$ (since in this case $T - 2N = r$).

Now, we proceed by induction on $q$: Suppose the statement holds for $(q-1)N + r$. That is, suppose

$$G_{[1,T-N]} = (1-\alpha)^{T-N-r}G_{[1,r]} + \sum_{i=0}^{q-2}(1-\alpha)^{iN}(1-\gamma)^{q-(i+1)}G_{[r+1,r+N]}$$

$$+ \sum_{i=1}^{q-2}\left(\sum_{j=0}^{i-1}(1-\gamma)^j(1-\alpha)^{(i-1-j)N}\right)\left((1-\gamma)\Delta_{[T-(i+1)N+1,T-iN]} + \gamma G_{[T-(i+1)N+1,T-iN]}\right)$$

Then, we observe the following identity if $i < q$:

$$G_{[T-iN+1,T-(i-1)N]} = (1-\gamma)\Delta_{[T-iN+1,T-(i-1)N]} + \gamma G_{[T-iN+1,T-(i-1)N]} + (1-\gamma)G_{[T-(i+1)N+1,T-iN]}$$

From this we can conclude:

$$G_{[T-iN+1,T-(i-1)N]} = (1-\gamma)^{q-i}G_{[T-qN+1,T-(q-1)N]}$$

$$+ \sum_{j=0}^{q-i-1}(1-\gamma)^j\left((1-\gamma)\Delta_{[T-(j+i)N+1,T-(j+i-1)N]} + \gamma G_{[T-(j+i)N+1,T-(j+i-1)N]}\right)$$

Now, put this together with (9):

$$G_{[1,T]} = (1-\alpha)^N G_{[1,T-N]} + (1-\gamma)^{q-1}G_{[T-qN+1,T-(q-1)N]}$$

$$+ \sum_{j=0}^{q-2}(1-\gamma)^j\left((1-\gamma)\Delta_{[T-(j+1)N+1,T-jN]} + \gamma G_{[T-(j+1)N+1,T-jN]}\right)$$

using the definition of $q, r$:

$$= (1-\alpha)^N G_{[1,T-N]} + (1-\gamma)^{q-1}G_{[r+1,r+N]}$$

$$+ \sum_{j=0}^{q-2}(1-\gamma)^j\left((1-\gamma)\Delta_{[T-(j+1)N+1,T-jN]} + \gamma G_{[T-(j+1)N+1,T-jN]}\right)$$

using the induction hypothesis:

$$= (1-\alpha)^{T-r}G_{[1,r]} + \sum_{i=0}^{q-2}(1-\alpha)^{(i+1)N}(1-\gamma)^{q-(i+2)}G_{[r+1,r+N]}$$

$$+ \sum_{i=1}^{q-2}\left(\sum_{j=0}^{i-1}(1-\gamma)^j(1-\alpha)^{(i-j)N}\right)\left((1-\gamma)\Delta_{[T-(i+1)N+1,T-iN]} + \gamma G_{[T-(i+1)N+1,T-iN]}\right)$$

$$+ (1-\gamma)^{q-1}G_{[r+1,r+N]}$$

$$+ \sum_{j=0}^{q-2}(1-\gamma)^j\left((1-\gamma)\Delta_{[T-(j+1)N+1,T-jN]} + \gamma G_{[T-(j+1)N+1,T-jN]}\right)$$

reindexing and combining the third line with the sum on the first line:

$$= (1-\alpha)^{T-r}G_{[1,r]} + \sum_{i=0}^{q-1}(1-\alpha)^{iN}(1-\gamma)^{q-(i+1)}G_{[r+1,r+N]}$$

$$+ \sum_{i=1}^{q-2}\left(\sum_{j=0}^{i-1}(1-\gamma)^j(1-\alpha)^{(i-j)N}\right)\left((1-\gamma)\Delta_{[T-(i+1)N+1,T-iN]} + \gamma G_{[T-(i+1)N+1,T-iN]}\right)$$

$$+ \sum_{j=0}^{q-2}(1-\gamma)^j\left((1-\gamma)\Delta_{[T-(j+1)N+1,T-jN]} + \gamma G_{[T-(j+1)N+1,T-jN]}\right)$$

reindexing:

$$= (1-\alpha)^{T-r}G_{[1,r]} + \sum_{i=0}^{q-1}(1-\alpha)^{iN}(1-\gamma)^{q-(i+1)}G_{[r+1,r+N]}$$

$$+ \sum_{i=2}^{q-1}\left(\sum_{j=0}^{i-2}(1-\gamma)^j(1-\alpha)^{(i-1-j)N}\right)\left((1-\gamma)\Delta_{[T-iN+1,T-(i-1)N]} + \gamma G_{[T-iN+1,T-(i-1)N]}\right)$$

$$+ \sum_{j=0}^{q-2}(1-\gamma)^j((1-\gamma)\Delta_{[T-(j+1)N+1,T-jN]} + \gamma G_{[T-(j+1)N+1,T-jN]})$$

$$= (1-\alpha)^{T-r}G_{[1,r]} + \sum_{i=0}^{q-1}(1-\alpha)^{iN}(1-\gamma)^{q-(i+1)}G_{[r+1,r+N]}$$

$$+ \sum_{i=2}^{q-1}\left(\sum_{j=0}^{i-2}(1-\gamma)^j(1-\alpha)^{(i-1-j)N}\right)\left((1-\gamma)\Delta_{[T-iN+1,T-(i-1)N]} + \gamma G_{[T-iN+1,T-(i-1)N]}\right)$$

$$+ \sum_{i=1}^{q-1}(1-\gamma)^{i-1}((1-\gamma)\Delta_{[T-iN+1,T-(i-1)N]} + \gamma G_{[T-iN+1,T-(i-1)N]})$$

$$= (1-\alpha)^{T-r}G_{[1,r]} + \sum_{i=0}^{q-1}(1-\alpha)^{iN}(1-\gamma)^{q-(i+1)}G_{[r+1,r+N]}$$

$$+ \sum_{i=2}^{q-1}\left(\sum_{j=0}^{i-1}(1-\gamma)^j(1-\alpha)^{(i-1-j)N}\right)\left((1-\gamma)\Delta_{[T-iN+1,T-(i-1)N]} + \gamma G_{[T-iN+1,T-(i-1)N]}\right)$$

which establishes the claim. $\qquad\square$

**Proposition 18.** *[Essentially Daniely et al. [2015], Lemma 5] The output of $S$ of $\mathrm{COMPOSE}(a,b)$ satisfies $|S| \leq 2(1+\lfloor\log_2(b-a+1)\rfloor)$. In the special case that $a=1$, $|S| \leq 1+\log_2(b)$. Moreover, each element of $\mathrm{COMPOSE}(a,b)$ is an interval of the form $[q2^k+1,(q+1)2^k]$ for some $q,k \in \mathbb{N}$, the intervals in $S$ are disjoint, and $[a,b] = \bigcup_{[x,y]\in S}[x,y]$.*