# OpenReview forum: "Momentum Aggregation for Private Non-convex ERM"
_NeurIPS.cc/2022/Conference — NeurIPS 2022 Accept_

### Official Review · Reviewer_nfxU · 2022-07-08

**Rating:** 5
**Confidence:** 4
**Soundness:** 3 good
**Presentation:** 2 fair
**Contribution:** 2 fair

**Summary:**

This paper proposes the use of momentum techniques in conjunction with the tree aggregation framework in differential privacy to obtain better rates for differentially private stationary points for the empirical loss. This problem has got recent attention in the literature, and all existing rates are slow ($O(d^{1/4}/\sqrt{N\epsilon})$) and require high running time. This paper focuses on both aspects, first providing algorithm with the above rate but with better running time, which is primarily based on noisy SGD with momentum (which crucially makes use of tree aggregation). And second, providing a sharper rate of $O((d/[N\epsilon]^2)^{1/3})$ for this problem (gradient complexity of the method is $O(N^{7/3}\epsilon^{4/3}/d^{2/3})$, but running time is actually at least cubic in the sample size). This is achieved by the use of a recursive gradient estimator, for which its terms have low sensitivity due to the small displacement of iterates, together with smoothness of the loss.

**Questions:**

1. Line 120:  The notation of the samples in the dataset from $x$  to $z$ is confusing, as $x$ already been used for data points. Maybe they should avoid summing intervals $[x,y]$ and prefer denoting intervals as $[y,z]$. This applies to other parts of the paper where this notation is used.
2. Page 4, line 143: an extra 'in' after m_t.
3. Page 4, equations after line 143. I find confusing that the sum in the second equality goes from $x+1$ to $y$. Does this mean that e.g., when $x=y$ we get an empty sum? BTW, it would be highly beneficial that, before unraveling the formula for $m_t$, the recursion $m_t=(1-\alpha)m_{t-1}+\alpha \nabla f(w)$ is introduced. Also there is a typo: w'_t should be w_{t'}.
4. Algoritm 1, page 5. It is never said (only in the appendix) what $S_i[1]$ is. Besides, the '$b-(S_i[1])$' I think should be '$t-(S_i[1])$'.
5. Algorithm 1, when generating the noise map $Z$, shouldn't it be $\chi_{[a2^b + 1 : (a+1)2^b]} and Z[a2^b + 1 : (a+1)2^b]$?
6. Page 5, line 157. $f_{[x,y)}$ should be $f_{[x,y]}$?
7. Page 5, line 161: $q$ is used to indicate the data point that is different on the neighboring datasets, but $q$ is already been used to indicate the epochs in the algorithms. This is not a mistake, but please consider a different notation. Other repetitions of use in notation (e.g. for $R$) are present throughout, which make reading quite confusing.
8. Page 7, Algorithm 2. $q_i$ is defined in the third line, but when it is actually used, it should be q_t instead of q_i, and indeed I think one could simply use '$q$' since It is defined by $t = qN + r$.
9. Page 7, Algorithm 2. I couldn't follow why there are two types of noises, depending on whether $t\leq 2N$ or larger. Is there a fundamental reason for this? I am also confused about the fact that the algorithm seems to rebuild all estimators from scratch every time they are needed, and then new noise is added to them. In this sense, it is not entirely clear to be that the privacy analysis here is accounting for all these repeated noise estimates. I would think it's easier and more natural to reuse the privatized terms from previous iterations, but possibly I am missing something here.
10. Page 7, line 196: should be an N instead of T in the rate?
11. Page 8, equation (2). Correct $G_{[1:t]}$ and $\alpha_{[1:t]}$ by $G_{[1,t]}$ and $\alpha_{[1,t]}$.
12. Page 8, equation (3). What is $r$ here? There could be multiple choices for it. Also, the notation is confusing, because of the use of $r_{[a,b]}$. I think it would be easier to understand the decomposition if they explicitly say that they are considering $t = qN + r$.
13. Equation (4) after line 551: Should Compose(t) be Compose[1,t] instead?
14. Line 472: $F_i$ should be $G_i$?
15. Line 492 and Appendix C in general: sometimes the bold notation of the functions $f_i$ is dropped.  Even in algorithm 5 f_1 is not written in bold but the rest iterations are.
16. Proof of lemma 11 (below line 564), in the first inequality there is a term $(1-\alpha)^t$ that should be $(1-\alpha)^{t-i}$. However, it is correct it in the next inequality.






**Limitations:**

Yes. Limitations on running time and possible additional improvements are discussed. It is not discussed though how far from tight the obtained upper bound is, which I think it is an important consideration.

**Strengths And Weaknesses:**

Strengths:

1. This is a definite improvement upon the state of the art.
2. The problem is of key relevance in differential privacy and  machine learning.
3. At the technical level, the algorithms do not require privacy amplification techniques, although they use multiple random shuffles of the data, which is somewhat baffling. Regarding this last point, I wonder whether the shuffling is necessary at all: I suspect this can be circumvented using analyses of incremental SGD, such as Nedic, Bertsekas (Incremental subgradient methods for nondifferentiable optimization, SIOPT 2001).

Weaknesses:

1. The improvements seem to be marginal, compared to existing literature.
2. Regarding algorithmic performance it is arguably more interesting to understand the out of sample performance of the algorithm.
3. The technique of recursive gradient estimation, used particularly in the second method, is already well known (see e.g., BGM'21 paper cited, and also Bassily, Guzman, Nandi 2021: "Non-Euclidean Differentially Private Stochastic Optimization"). The paper does not reflect this fact in its presentation. The main difference between the current submission and past work is that in the current paper this is analyzed for multiple passes on the data, which adds another layer of complexity into the analysis.
4. The paper contains typos and some small inconsistencies which make reading difficult.

---

> ### Author Response · Authors · 2022-08-02
> **Response to Reviewer nfxU**
>
> Thank you for your comments! Your comments are very helpful for us to improve the quality of our paper. We will address some of your concerns below.
>
> Response to weaknesses:
>
> 1. We want to emphasize that achieving the improved convergence bound of $\tilde O(d^{1/3}/(\epsilon N)^{2/3})$ that we did is not an easy task. The bound $O(d^{1/4}/(\sqrt{\epsilon N})$ is achieved way back in 2017 by [1] but the bound hasn’t been improved since then despite the progress in multiple privacy amplification techniques (shuffling, sampling). That is, we obtain a significant asymptotic improvement over the state of the art.
>
> 2. Thanks for the suggestion! It’s actually quite straightforward to convert our bound to an out-of-sample bound. If we use the result from [3], our out-of-sample bound would be roughly $\tilde O(d^{1/3}/(\epsilon N)^{2/3}+ \epsilon)$. By optimizing over $\epsilon$, the out-of-sample bound will become $\tilde O(1/N^{2/5})$ (thus it would compare favorably to the existing literature as well since the best bound that we are aware of is [4]). We will add a small discussion on this in the final revision for interested readers.
>
> 3. Thanks for the pointer, we will add it to our citation! The reviewer is correct that variance reduction is not a new technique. However, applying this technique on sgd with momentum across multiple epochs (and with shuffling) is definitely not trivial. And as we have pointed out in the paper, just naively using variance reduction will not achieve the convergence bound in our second algorithm.  [2] also uses variance reduction across multiple epochs but they can’t improve on the previous $\tilde O(d^{1/4}/\sqrt{\epsilon N})$ bound. Our improved rate does not come from reducing *variance*, but from reducing *sensitivity*. Unlike standard variance reduction, it is privacy-specific: there is no advantage in using our analysis for non-private optimization.
>
> Response to questions: Thanks very much for your very careful and insightful review. We also appreciate your detailed line comments. We will address all of them to improve the final copy. Below we respond to a few of your more specific questions:
>
> 3. Sorry about that, the sum should go from x->y so there will not any empty sum.
>
>  9. Notice that in our decomposition, in the first 2 epochs, we don’t have the extra parameter $\gamma$ that helps us control the noise added to the momentum. Thus if we just naively add noise based on the maximum sensitivity (which is $V_{>2N}$), the noise will blow up since it scales with T/N (it would give us the same bound as the first method). Thus, by dividing the sensitivity into 2 cases, we can make the momentum of the first 2 epochs $(\epsilon,\delta)$-DP and every other epochs $(\epsilon,\delta)$-dp without adding too much noise. Overall, the whole procedure would be $(O(\epsilon), O(\delta))-$DP. In terms of aggregation function in the second algorithm, we admit that it looks quite confusing but in its essence, it’s still tree-based aggregation similar to the first method. Thus, the behavior of the repeated noise would be the same so we can still apply the same privacy analysis.
>
> 10. Yes, it should be since we are deriving bound with respect to the total number of samples.
>
> 12. Yes, r is the remainder of t/N as you said. We will make sure to clarify this in the final revision.
>
> 15. Yes, whenever we write Compose(t), it means Compose(1,t).
>
> [1] Jiaqi Zhang, Kai Zheng, Wenlong Mou, and Liwei Wang. Efficient private erm for smooth objectives. In 397 Proceedings of the 26th International Joint Conference on Artificial Intelligence, pages 3922–3928, 2017.
>
> [2] Di Wang, Minwei Ye, and Jinhui Xu. Differentially private empirical risk minimization revisited: Faster and 388 more general. arXiv preprint arXiv:1802.05251, 2018.
>
> [3] Jung, Christopher, et al. "A new analysis of differential privacy's generalization guarantees." arXiv preprint arXiv:1909.03577 (2019).
>
> [4] Zhou, Yingxue, et al. "Private stochastic non-convex optimization: Adaptive algorithms and tighter generalization bounds." arXiv preprint arXiv:2006.13501 (2020).

---

> > ### Comment · Reviewer_nfxU · 2022-08-06
> > **Comments on authors' response**
> >
> > I would like to thank the authors for addressing my comments. I am maintaining my score, as I feel it is justified under the following considerations.
> >
> > 1. I agree that obtaining these improvements is not easy task. My comment referred exclusively to the marginal improvement (from $1/2$ exponent to $2/3$ exponent). This in fact would be a great result if one could additionally prove that no further improvements are possible (e.g., by providing a lower bound), but nothing of the sort is discussed in the paper. In summary, the paper provides a definite improvement, yet it does not resolve the question of optimal rates in differentially private stationary points (and I discussed fairly about both aspects in my review).
> >
> > 2. Yes, I agree it is possible to obtain generalization bounds through differential privacy, but this reduction is typically not for free. In fact, using the authors' claimed generalization bound, the optimization of $\epsilon$ would lead to norm of the gradient $\tilde O(d^{1/5}/N^{2/5})$, and not $\tilde O(1/N^{2/5})$, as claimed. In fact, doing this analysis for general values of $\epsilon\in(0,1]$ leads to $O((\sqrt{d}/[N\epsilon])^{2/5})$ bounds. One can see here the polynomial degradation of the accuracy bound between empirical and population loss. I suspect that investigation of population guarantees in this case must go beyond these reductions.
> >
> > Above I was simply using the claimed bound by the authors. But in fact I am not quite sure this is the right generalization bound. The cited paper only considers 1 dimensional statistical queries, but when dealing with vector valued queries, the generalization bound in terms of norms should lead to an $\tilde O(\sqrt{d}\epsilon)$ additive term. The bounds in this case would have a much worse dependence on dimension. Can the authors please elaborate on this?
> >
> > 3. I agree that the way sensitivity control and privacy analysis of this paper is quite interesting and novel. And I suspect the improvements of the paper would not be possible without these ideas (so here I agree with the authors).
> >
> > 4. The writing of the paper is suboptimal, which is evidenced in the numerous typos, confusing notation, and other issues I described in my review. I appreciate the authors' receptive attitude to these comments, but I cannot judge how the corrected version of the paper will be, and in its current form I don't think it is suitable for a top venue such as NeurIPS.

---

> > > ### Author Response · Authors · 2022-08-08
> > > **Response to Reviewer nfxU**
> > >
> > > We appreciate the reviewer for the insightful discussion! We will respond to your questions below.
> > >
> > > 1. Regarding the lower bounds: as you are no doubt aware, lower bounds are notoriously harder to show than upper bounds. In fact, even for *non-private* stochastic non-convex optimization, the optimal lower bounds are only a few years old [5]. We think it might be unfair to characterize our asymptotic improvement as marginal. Again, by analogy to non-private non-convex optimization, prior to the lower bound there was a string of relatively high-impact papers improving the bounds from $O(1/N^{1/4})$, to $O(1/N^{3/10})$ [Reddi et al. 2016]  to $O(1/N^{1/3})$ [Fang et al 2018] before it was established that $O(1/N^{1/3})$ was optimal. We submit that our work has a similar flavor, but we will add a discussion of this limitation to the limitations section.
> > >
> > > 2. Sorry for the confusion, the approximate generalization bound is indeed not $\tilde O(1/N^{2/5})$ like you said. We omit most of the details in the previous answer just to illustrate the point that it is easy to approximately convert our bound to out-of-sample bound due to the generalization properties of differential privacy. Technically, using Theorem 16 of [3], the generalized error would be bounded by $\tilde O(d^{1/3}/(\epsilon N)^{2/3} + \sqrt{d}\epsilon\alpha N)$ where $\alpha$ comes from the sensitivity of our momentum queries. Since $\alpha = 1/N$ (based on the analysis of Theorem 9 in our paper), by optimizing over $\epsilon$, we will have the bound $\tilde O(d^{2/5}/N^{2/5})$. So the reviewer is correct that this can be a problem with high-dimensional settings. We agree with the reviewer that this black-box ERM to population approach may not be optimal and that some further techniques might be necessary. We hope that our work provides a pathway towards such a final optimal result.
> > >
> > > 3. We’ve attached a revised version of our paper above. We have addressed the most important issues which are the typos for now. Hopefully, this helps improve the readability of our paper. We will also add more clarifying discussions of our results and techniques to the final revision of our paper.
> > >
> > > [5] Arjevani, Yossi, et al. "Lower bounds for non-convex stochastic optimization." arXiv preprint arXiv:1912.02365 (2019).

---

> > > > ### Comment · Reviewer_nfxU · 2022-08-09
> > > > **Final comments**
> > > >
> > > > 1. Thank you. Putting the results in perspective would be nice. Proving lower bounds in this case is indeed difficult, and at least the paper should comment on this aspect.
> > > >
> > > > 2. Thank you for the clarification.
> > > >
> > > > 3. Thank you. I can see some notational improvements, yet I still have some major concerns about the privacy analysis. I will reiterate question 9 I left in the first review:
> > > >
> > > > "I am also confused about the fact that the algorithm seems to rebuild all estimators from scratch every time they are needed, and then new noise is added to them. In this sense, it is not entirely clear to be that the privacy analysis here is accounting for all these repeated noise estimates. I would think it's easier and more natural to reuse the privatized terms from previous iterations, but possibly I am missing something here."
> > > >
> > > > It seems the authors partially answer this in their response (with a different numbering?), but the paper update does not seem to clarify this issue.

---

### Official Review · Reviewer_vQgL · 2022-07-11

**Rating:** 6
**Confidence:** 3
**Soundness:** 3 good
**Presentation:** 2 fair
**Contribution:** 3 good

**Summary:**

The paper proposed a few new algorithms for private ERM. The key idea of the new algorithms is using tree aggregation in momentum normalized SGD from Cutkosky and Mehta, 2020. The proposed algorithms can achieve better convergence rate than the existing method.

**Questions:**

The results looks quite solid to me. One question I have is why to use normalized SGD as the base algorithm, could the authors elaborate more on it?

**Limitations:**

Limitations are well-discussed.

**Strengths And Weaknesses:**

Strengths:

1. The idea of using tree aggregation for momentum seems to be quite novel in the field.
2. The proposed algorithms can achieve better convergence rate.

Weaknesses:
1. The proposed algorithm seems to require a larger memory to store some tree aggregation nodes.

---

> ### Author Response · Authors · 2022-08-02
> **Response to Reviewer vQgL**
>
> Thank you for your comments! Your comments are very helpful for us to improve the quality of our paper. We will address some of your concerns below.
>
> Response to weaknesses:
>
> We do require a bit more memory than just normal SGD since we need $O(\log N)$ nodes to store the tree. Note however that we do not need to store the entire tree, just one node at each of the $\log(N)$ heights. If memory is a significant issue, we can likely tradeoff this storage requirement for an $O(\log N)$ time factor by recomputing the noise for each tree node on-the-fly every time it is needed using a PRNG.
>
> Response to questions:
>
> The main reason to use normalized sgd is that the analysis of normalized sgd with momentum is a bit more simple than that of sgd with momentum (since each update now has a precisely controlled displacement). We suspect that a similar result holds with an unnormalized update, but since our analysis with tree-aggregated momentum is already quite involved, we don’t want to make the analysis even more complicated.

---

### Official Review · Reviewer_85uj · 2022-07-18

**Rating:** 7
**Confidence:** 3
**Soundness:** 3 good
**Presentation:** 3 good
**Contribution:** 3 good

**Summary:**

The paper proposes a new privacy-preserving algorithm and proves convergence guarantees on the existing differentially private normalized SGD algorithm and the variant proposed in the paper. The new variant of sensitivity reduced normalized sgd achieves better convergence guarantees on smooth non-convex problems than previous algorithms. The improvement comes from using tree-aggregation technique that reduces the amount of noise needed to make partial sums differentially private.

**Questions:**

Lines 103-111: Since the noise is being added to each gradient to make each of them differentially private, and the gradients in an epoch come from different samples, even after summing the gradients in an epoch, I think the differential privacy for each sample will remain  $(\epsilon/\sqrt{t}, \delta/t)$. I think the composition rule applies when the gradients are from the same sample, but here since a sample will not be seen until the next epoch, the differential privacy guarantee does not reduce. Is the paragraph explaining the case when gradients are coming from the same sample (that is, GD instead of SGD)? Can the authors clarify this?


Typo in Algorithm 1: step size is missing in the statement $w_{t+1} ← w_t − m_t/\|m_t\|$

**Limitations:**

.

**Strengths And Weaknesses:**

Strengths:
1. The paper uses the tree-aggregation technique to propose a new technique that achieves better error guarantee on smooth, non-convex objectives than prior works (although the dependence on the dimension $d$ and $\epsilon$ is worse).
2. The paper presents a new analysis of differentially private normalized sgd which achieves same error rates as previous papers, without needing any privacy amplification techniches or needing full batch gradient descent which were needed in previous papers.


Weaknesses:
1. Although the error dependence of algorithm 2 on N is better, the dependence on d and $\epsilon$ is worse for the new algorithm.
2. The paper admits that runtime is worse for the new algorithm is proportional to $N^{11/3}/d^{2/3}$. Even in the regime when $N$ is proportional to $d$, this is $N^3$. Also, number of steps needed by algorithm 1 is proportional to $N^2$. Both of these numbers are large, but the $N^3$ runtime of the new algorithm is particularly problematic.
3. The paper admits that it is not obvious that privacy amplification by shuffling can be directly applied to their tree-aggregation based method.
4. The paper does not have any experiments showing the convergence properties of the new algorithm. I understand that this is a theoretical paper, but some small scale experiments would be nice to better gauge the algorithm.

---

> ### Author Response · Authors · 2022-08-02
> **Response to Reviewer 85uj**
>
> Thank you for your comments! Your comments are very helpful for us to improve the quality of our paper. We will address some of your concerns below.
>
> Response to weaknesses:
> 1. You are correct that the dependence on $d$ and $\epsilon$ is worse in the new bound. However, $\epsilon$ is usually a small constant [1] so the dominant factor in the denominator would be the number of samples N. In regards to dependence on $d$, it would be interesting to check if we are able to get the “best of both worlds” bound $\tilde O(d^{1/4}/(\epsilon N)^{2/3})$. However, we conjecture that $d^{1/3}$ is the best that we can do for $O(1/N^{2/3})$ bound since the dependence on $d$ tends to get worse when the dependence on $N$ gets better.
>
> 2. We admit that the runtime is not ideal. However, we want to emphasize that the runtime of our first algorithm is not particularly large compared to previous work. Most of the previous algorithms also have quadratic runtime similar to our first algorithm. The only algorithm that has sub-quadratic runtime is DPSRM with the runtime of $O(N^{3/2})$. However, they require the batch size to be $O(\sqrt{N})$ which we would argue might be an even bigger problem in practical applications. For our second algorithm, our main goal is to design an algorithm that achieves a new SOTA convergence rate for non-convex ERM (which we did achieve with the convergence rate of $\tilde O(d^{1/3}/(\epsilon N)^{2/3}))$. We suspect that we can reduce the runtime by aggregating the momentum in a more clever way but we will leave this for future work. The main focus of this work is to demonstrate that achieving a bound that is better than $\tilde O(1/\sqrt{N})$ is possible, which previous methods can’t achieve even with infinite computations.
>
> 3,4. Thanks for the suggestions
>
> Response to questions:
>
> Thanks for the question! You are correct that the privacy guarantee will not degrade if each gradient is completely unrelated to the other. We meant to add the noise of standard deviation $O(\sqrt{T}/\epsilon)$ to each sum $s$ instead of each $g_t$, which would require advanced composition. Of course, if we instead add noise of $O(1/\epsilon)$ to each $g_t$ this does not need to use composition, but will still result in $O(\sqrt{T}/\epsilon)$ error in each partial sum. The $\sqrt{T}$ penalty is reduced to a logarithmic term by the tree aggregation technique.
>
>
> [1] Du, Jian, et al. "Dynamic differential-privacy preserving sgd." arXiv preprint arXiv:2111.00173 (2021).

---

### Meta-Review · Area_Chair_GNCU · 2022-08-26

**Recommendation:** Accept
**Confidence:** Less certain

**Metareview:**

The authors obtain improved guarantees for differentially private smooth, non-convex empirical risk minimization. Following the discussion stage, there appears to be a consensus in favor of accepting this paper, due largely to the clear theoretical improvement and the fundamental nature of the problem. However, reviewer nfxU, who remains unenthusiastic about the paper, is still concerned about lack of clarity regarding a number of algorithm design choices: the necessity of making the entire optimization trajectory DP, whether or not the magnitude of noise changes between steps, and whether shuffling is the data is truly necessary. While these points do not cause direct concern over the correctness of the results, they are nevertheless important to clarify, and I urge the authors to do so thoroughly in the camera-ready revision.

**Award:**

No

---

### Decision · Program_Chairs · 2022-09-14

Accept